# The functional significance of the RPA- and PCNA-dependent recruitment of Pif1 to DNA

Oleksii Kotenko ⬤ & Svetlana Makovets ⬤ ✉

## Abstract

Pif1 family helicases are multifunctional proteins conserved in eukaryotes, from yeast to humans. They are important for the genome maintenance in both nuclei and mitochondria, where they have been implicated in Okazaki fragment processing, replication fork progression and termination, telomerase regulation and DNA repair. While the Pif1 helicase activity is readily detectable on naked nucleic acids in vitro, the in vivo functions rely on recruitment to DNA. We identify the single-stranded DNA binding protein complex RPA as the major recruiter of Pif1 in budding yeast, in addition to the previously reported Pif1-PCNA interaction. The two modes of the Pif1 recruitment act independently during telomerase inhibition, as the mutations in the Pif1 motifs disrupting either of the recruitment pathways act additively. In contrast, both recruitment mechanisms are essential for the replication-related roles of Pif1 at conventional forks and during the repair by break-induced replication. We propose a molecular model where RPA and PCNA provide a double anchoring of Pif1 at replication forks, which is essential for the Pif1 functions related to the fork movement.

**Keywords** DNA Repair; DNA Replication; Pif1 Recruitment; RPA; Telomerase
**Subject Category** DNA Replication, Recombination & Repair

## Introduction

Genome maintenance involves a plethora of different enzymes, such as DNA polymerases, helicases, nucleases, etc. In many cases, these enzymes share the same enzymatic activity in vitro. Why have the cell evolved multiple proteins to encode similar biochemical activities? Perhaps, because the same enzymatic activity might be requited for different molecular functions. For example, both Sgs1 (BLM in mammals) and Srs2 are 3'→5' helicases in budding yeast. Sgs1 operates as part of the Sgs1/Top3/Rmi1 complex, which suppresses excessive recombination by disrupting the strand invasion, and also cooperates with Dna2 to resect 5' ends during DNA break repair (Myung et al, 2001; Cejka et al, 2010; Zhu et al, 2008). In contrast, Srs2 functions on its own and regulates the recombination machinery through dislodging Rad51 from single-stranded DNA (ssDNA) (Veaute et al, 2003; Krejci et al, 2003). This is important for inhibiting excessive recombination at replication forks (Papouli et al, 2005; Pfander et al, 2005; Burgess et al, 2009) and also for promoting resynthesis of resected DNA during repair (Vasianovich et al, 2017).

While the catalytic domains of similar enzymes share significant homology, the sequences outside of these domains are very different. The latter often define protein–protein interactions, which can make an enzyme a part of a protein complex with a distinct function, as in the case of Sgs1. Alternatively, the non-catalytic regions of a protein could be required for the efficient protein recruitment to specific DNA loci and its retention there during its activity. Therefore, the regions outside of the catalytic domains can define the functional specificity of the proteins and determine their substrates and molecular roles in DNA metabolism.

PCNA and RPA are the best-known recruiters of the genome maintenance machinery to DNA, mainly to the sites of replication and repair. PCNA is a homo-trimer organised into a ring which encircles the DNA and anchors the components of the replication machinery and other DNA-operating proteins (Krishna et al, 1994). Many of these contain PCNA-Interacting Peptides (PIPs) which bind one of the PCNA monomers in order to stabilise the enzymes on DNA (Gulbis et al, 1996; Zheng et al, 2020; Lancey et al, 2020). PCNA can be SUMOylated or ubiquitinated in response to replication fork stalling and this promotes the recruitment of specialised enzymes which then deal with stalled replication forks (Hoege et al, 2002; Stelter and Ulrich, 2003; Armstrong et al, 2012). SUMO and ubiquitin attached to PCNA provide further protein–protein interaction points for the enzymes which in addition to PIPs have SUMO-Interacting Motifs (SIMs) and Ubiquitin-Interacting Motifs (UIMs).

RPA is a heterotrimer which binds ssDNA and inhibits the DNA secondary structure formation. RPA is also involved in the recruitment of other proteins to DNA. Accumulation of ssDNA either at DNA breaks, due to the end resection, or at replication forks as a result of blocked DNA synthesis leads to excessive RPA accumulation. This in turn is sensed as DNA damage by the damage-signalling machinery, because it allows sufficient co-localisation of the damage sensors on ssDNA through their binding to RPA (Bonilla et al, 2008; Deshpande et al, 2017; Rouse and Jackson, 2002; Zou and Elledge, 2003). RPA can also recruit DNA processing enzymes, for example the BTR (BLM-TOP3A-RMI) complex and PRIMPOL in mammals (Shorrocks et al, 2021; Guilliam et al, 2017), as well as Bre1 and Dna2 in yeast (Acharya et al, 2021; Liu et al, 2021). Most of the known

Institute of Cell Biology, University of Edinburgh, King's Buildings, Alexander Crum Brown Road, Edinburgh EH9 3FF, UK. ✉E-mail: smakovet@staffmail.ed.ac.uk

RPA-interacting partners bind the N-terminal domain of the largest subunit of the complex, Rfa1 in yeast and RPA70 in mammals. These partners share the amino acid sequence similarity at the RPA-Binding Motifs (RBMs) which have been recently emerging in mammals, but not in yeast (Shorrocks et al, 2021). Therefore, both RPA and PCNA recruit proteins to DNA through binding to short motifs in their interacting partners.

The Pif1 family helicases are part of the DNA maintenance machinery in eukaryotes. They have a number of functions in DNA replication and repair. *PIF1* is not essential in mice (Snow et al, 2007) but mutations in the human gene have been associated with cancer (Gagou et al, 2011; Gagou et al, 2014; Li et al, 2021). The budding yeast *Saccharomyces cerevisiae* have two Pif1 family helicases, Pif1 and Rrm3 (Ivessa et al, 2000). During replication, they help the forks to progress through the hard-to-replicate regions (such as telomeres, centromeres, tRNA genes, etc) and aid replication termination (Makovets et al, 2004; Ivessa et al, 2000; Osmundson et al, 2017; Tran et al, 2017; Chen et al, 2019; Deegan et al, 2019). While Rrm3 has been found travelling with the replication forks (Azvolinsky et al, 2006), Pif1 localises only to certain loci in the nuclear genome (Tran et al, 2017; Phillips et al, 2015; Chen et al, 2019) and unlike Rrm3, it is also required for the mitochondrial genome maintenance (Foury and Kolodynski, 1983). In addition, Pif1 is present at DNA breaks (Wagner et al, 2006; Makovets and Blackburn, 2009), where it inhibits erroneous break repair by de novo telomere addition (DNTA) (Schulz and Zakian, 1994). Similarly, Pif1 inhibits telomerase at telomeres as the lack of Pif1 results in increased telomere length (Schulz and Zakian, 1994). Another role of Pif1 is in double-strand break (DSB) repair by break-induced replication (BIR) where it is implicated in assisting the fork movement during the repair (Wilson et al, 2013). This function requires the interaction of Pif1 with PCNA via a PIP located in the C-terminus of Pif1 (Buzovetsky et al, 2017). Here, we report that RPA is the major recruiter of Pif1 to DNA, identify an RBM in the N-terminus of Pif1 and analyse the functional significance of both RPA- and PCNA-dependent recruitment of Pif1 to DNA.

## Results and discussion

### Pif1 interacts with RPA via an N-terminally located conserved motif

The *PIF1* gene in the budding yeast *S. cerevisiae* codes for both nuclear and mitochondrial versions of the protein, depending on whether the translation starts from the first methionine codon (M1, produces mitochondrial Pif1) or from the second one (M40, produces nuclear Pif1). As before (Makovets and Blackburn, 2009; Vasianovich et al, 2014), in this work we had two different alleles placed at the *PIF1* locus, *pif1-m2* (the M40 codon mutated) which encoded only mitochondrial Pif1, and *pif1-m1* (the M1 is mutated) for the expression of the nuclear Pif1. All the in vivo manipulations in this work, such as mutations and protein tagging, involved only the nuclear allele.

Pif1 has been recently reported to interact with RPA (Maestroni et al, 2020). It consists of the centrally located helicase domain surrounded by unstructured N- and C-termini (Lu et al, 2018). To test if either the N- or the C-terminus of Pif1 might interact

with RPA, recombinant Pif1N(42–250) and Pif1C(750–859) protein fragments, corresponding to the N- and C-termini of the nuclear isoform of yeast Pif1, were expressed in *Escherichia coli* as GST-fusions, immobilised onto glutathione beads and used for pulldowns from yeast cell lysates. The pull-down samples were then resolved using SDS-PAGE and probed for yeast RPA by western blotting (Fig. 1A). Rfa1, the largest subunit of the RPA complex, was detected in the samples corresponding to GST-Pif1N(42–250), but neither GST-Pif1C(750–859) nor the GST only control (Fig. 1A,B). Therefore, the N-terminus of Pif1 can interact with RPA.

A number of different proteins are known to be recruited to DNA through the interaction with the N-terminal OB domain of Rfa1 in *S. cerevi*siae or its mammalian homologue RPA70 (Bochkareva et al, 2005; Lin et al, 1996; Xu et al, 2008; Ball et al, 2007; Olson et al, 2007; Acharya et al, 2021; Liu et al, 2021; Shorrocks et al, 2021). Rfa1 shares a substantial structural similarity with RPA70 (Bochkareva et al, 2005; Seeber et al, 2016). However, *S. cerevi*siae Rfa1N so far has been co-crystallised only with a peptide from *Kluyveromyces lacti*s Ddc2, revealing the molecular details of the potential RBM in budding yeast (Deshpande et al, 2017). We analysed Pif1N(42–250) for any sequence similarities to the *K. lactis* Ddc2 peptide (Fig. 1C, top). The short LDLL stretch in Pif1 was similar to LELV in the Ddc2 peptide. The presence of the negatively charged residues E64, D65, D71, D73, and D74 flanking this four amino acid stretch in Pif1 was consistent with how both RPA70 and Rfa1 interact with their partner proteins. These negatively charged residues normally interact with the lysines on RPA70/Rfa1 which are positioned on both sides of the hydrophobic cleft contacting the hydrophobic cores of the known RBMs (Bochkareva et al, 2005; Guilliam et al, 2017; Deshpande et al, 2017; Zhou et al, 2015).

To test if this hypothetical motif was required for the interaction with the N-terminus of Rfa1, we first constructed a GST-fusion with a shorter Pif1N fragment Pif1N(56–77) which contained the motif, and then made its derivative with the LDLL sequence mutated to four alanines (Fig. 1C, bottom). In order to test if these constructs could interact with the N-terminal domain of Rfa1, we expressed Rfa1N(1–132)-4myc in *E. coli*. Rfa1N-4myc was pulled down using GST-Pif1N(56–77) as a bait and this interaction was significantly reduced by the LDLL→AAAA (Pif1-rbm) substitution (Fig. 1D,E, lanes 1–3). Similar to Pif1N(56–77), Pif1N(42–250) could bind Rfa1N-4myc and LDLL→AAAA affected the interaction in the context of the full-length Pif1 N-terminus too (Fig. 1D,E, lanes 1, 4–5). Therefore, the interaction between Pif1 and RPA is direct and occurs via the N-terminally located RBM motif in Pif1 and the N-terminal OB fold domain of Rfa1.

### Both RBM and PIP motifs are involved in the Pif1 recruitment to DSBs and chromosome termini

Pif1 inhibits telomerase at DNA ends, both DSBs and telomeres (Schulz and Zakian, 1994). It is known to localise to DSBs (Wagner et al, 2006; Makovets and Blackburn, 2009). We hypothesised that either the newly established Pif1-RPA interaction or the previously found binding of Pif1 to PCNA might contribute to the recruitment of Pif1 to DNA breaks. To address these hypotheses, the Pif1 recruitment to a single inducible DSB was analysed by CHIP-qPCR in a set of isogenic strains producing different versions of nuclear

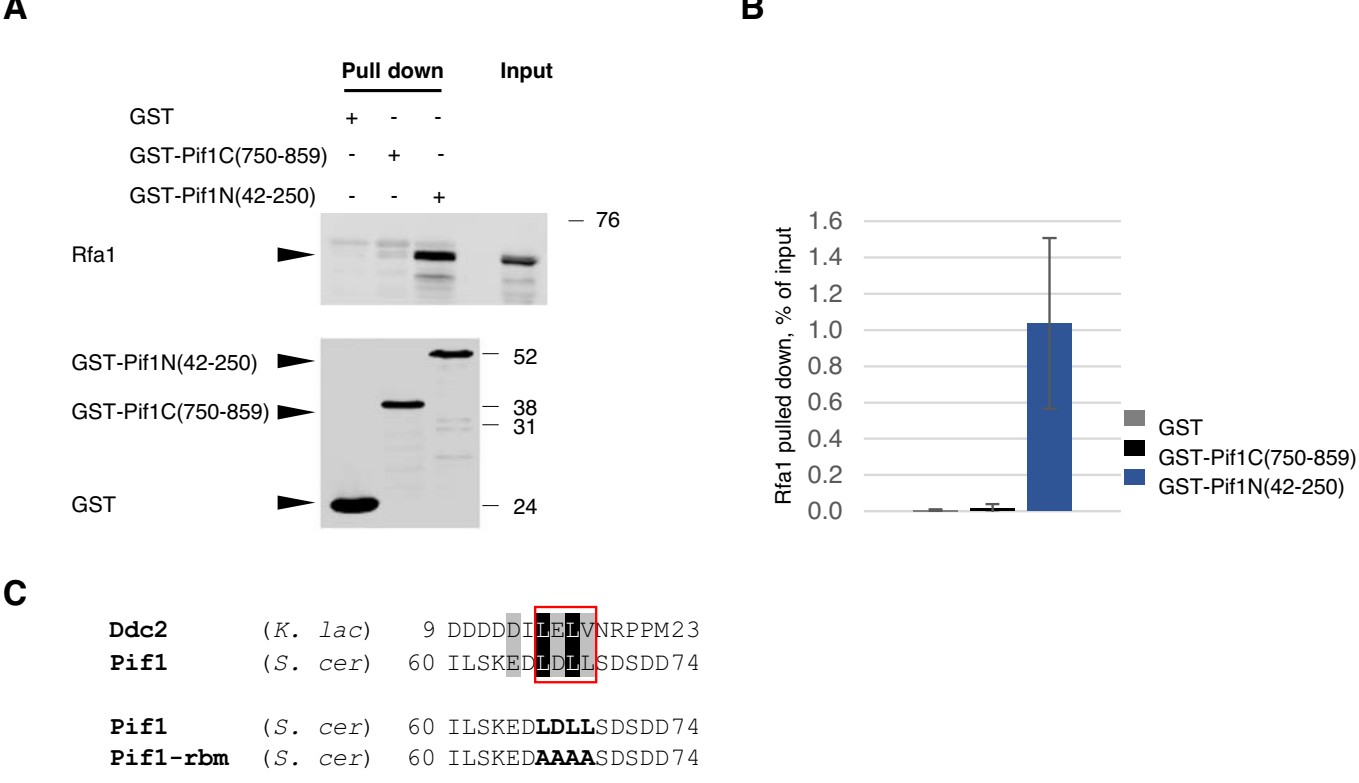

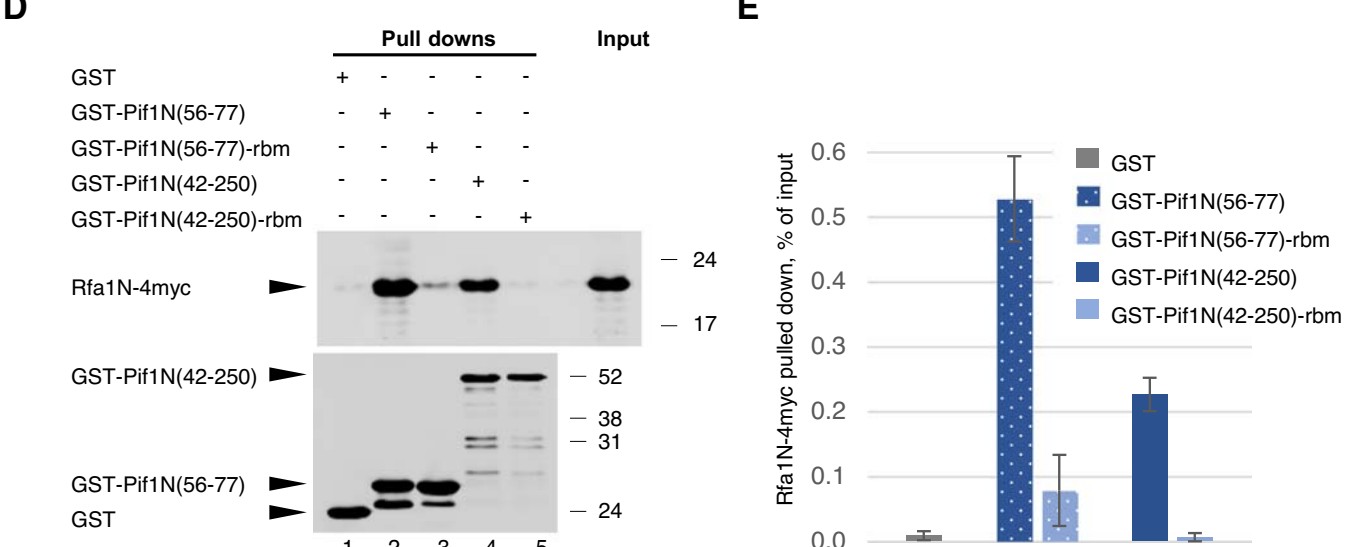

Pif1-4myc: the wild-type Pif1 and the three mutant derivatives deficient in their interaction with RPA (Pif1-rbm), PCNA (Pif1-pip) or both (Pif1-rbm-pip).

As expected, Pif1-4myc was readily detected at the HO-induced DSBs, while Pif1-rbm-4myc showed a drastically reduced localisation (Fig. 2A) which could not be attributed to any changes in the protein levels of the mutated proteins (Fig. EV1). The localisation of Pif1-pip-4myc to the DSBs was compromised the

least, suggesting that the RBM motif was more important for the Pif1 recruitment to DSBs than the PIP. The compromised localisation of Pif1-pip-4myc to the DSBs was not due to a non-specific effect of the *pip* mutation on the interaction between Pif1 and RPA, because this mutation did not affect the Pif1-RPA interaction tested by co-immunoprecipitation (Fig. EV2). The Pif1-rbm-pip-4myc derivative with both the RBM and the PIP motifs mutated showed a further decrease in the localisation to DSBs in

**Figure 1.   Pif1 interacts with RPA.**

(**A**) Proteins recovered after a pulldown from yeast cell lysates using recombinant GST-Pif1C and GST-Pif1N were analysed by western blotting. Rfa1 was detected using anti-RPA (*S. cerevisiae*) antibody (top image) and the baits in the pull-down samples were analysed using anti-GST antibody (bottom image). Here and in (**D**), the numbers on the right indicate the molecular weight of the size marker proteins (in kDa) run alongside the experimental samples. (**B**) Quantification of the experiments in (**A**). The plots show average $+/-$ SD ($n = 3$ biological replicates). (**C**) Sequence alignment of the previously identified Rfa1-interacting peptide of Ddc2 from *K. lactis* with the putative RBM motif in Pif1 (top). The amino acids highlighted in black are identical, the ones highlighted in grey are similar. The red box shows the conserved hydrophobic core in the RBM motifs. The mutated version of the RBM in Pif1 (Pif1-rbm) is shown at the bottom, with the substituted amino acids in bold. (**D**) Analysis of the interactions between GST-Pif1N and its GST-Pif1N-rbm derivative with Rfa1N-4myc using pulldowns followed by myc-specific western blotting (top image). The bottom image shows GST-fusion proteins in the pull-down samples analysed using anti-GST antibody. (**E**) Quantification of the experiments in (**D**). The plots show average $+/-$ SD ($n = 3$ biological replicates). Source data are available online for this figure.

comparison with either of the single mutation derivatives, suggesting that Pif1 is recruited to DSBs by two independent modes. One of them relies on RPA and the other one is PCNA-dependent. Based on the mutant phenotypes, the RPA-dependent recruitment plays a bigger role in the Pif1 localisation to DSBs, either due to a stronger protein–protein interaction or due to a higher abundance of RPA at resected breaks.

Next, we tested if Pif1 localisation to the chromosome termini of unchallenged cells also requires the RBM and PIP motifs. By using qPCR primers specific to the Y' sequence adjacent to the telomeric repeats, we were able to detect Pif1-4myc enrichment at the Y'-telomeres, while both Pif1-rbm-4myc and Pif1-rbm-pip-4myc showed compromised recruitment (Fig. 2B). Although we could not detect a statistically significant effect of the *pip* mutation either in the presence or absence of the functional RBM, the *pip* mutation may still cause a mild effect, similar to how it was observed for the Pif1 recruitment to DSB (Fig. 2A).

## The RBM and PIP motifs in Pif1 are required for its role in telomerase inhibition

Pif1 has a well-established role at DSBs, where it prevents erroneous break repair by DNTA. Therefore, the interactions with RPA and PCNA required for the Pif1 localisation to DSBs might be important for the suppression of the break healing by telomerase. To address this hypothesis, we used the previously described genetic system allowing to evaluate DSB repair by DNTA (Makovets and Blackburn, 2009), as outlined in Fig. EV3A. Both the *rbm* and the *pip* mutations affected the ability of Pif1 to inhibit telomerase at DSBs (Fig. 2C). In the cells with Pif1-rbm, the frequency of the G418$^S$ Ura$^-$ colonies formed after the DSB induction was increased compared to the Pif1-WT control strain, but it was noticeably lower than in the cells lacking nuclear Pif1. This hypomorphic phenotype was consistent with the partial defect in the Pif1-rbm localisation to DSBs (Fig. 2A). The cells with Pif1-pip had the frequency of the G418$^S$ Ura$^-$ colonies similar to the control strain, suggesting that the loss of the PIP motif had a minor if any effect on the telomerase inhibition at the breaks. However, the strain with Pif1-rbm-pip had a higher frequency of the G418$^S$ Ura$^-$ colonies than either of the single motif mutants (Fig. 2C), thereby unmasking the significance of the PIP motif in the telomerase inhibition at DNA breaks.

This genetic system is known to over-estimate the DNTA values in the cells with functional Pif1 as the low frequency of G418$^S$ Ura$^-$ colonies in the Pif1-proficient cells stems from low-frequency recombination elsewhere in the genome (Makovets and Blackburn, 2009). To overcome this problem, the G418$^S$ Ura$^-$

colonies were analysed by Southern blotting to screen for the actual DNTA events (Fig. EV3B). As expected, the G418$^S$ Ura$^-$ colonies from the Pif1-deficient strains contained de novo telomeres adjacent to the break locus, while the Pif1-proficient G418$^S$ Ura$^-$ colonies were all false-positive for newly acquired telomeres (Fig. 2D). Most of the colonies formed by the cells with Pif1-rbm, 27 out of 35, had de novo telomeres. The mutants with Pif1-pip showed a significantly smaller fraction of DNTA events among the analysed colonies, only 10 out of 56, but noticeably different from the wild-type cells. Therefore, Pif1-pip is also only partially functional in inhibiting telomerase at DSBs. But it is a weaker hypomorph than Pif1-rbm and the two defects are additive when combined in Pif1-rbm-pip (Fig. 2C). We conclude that both RBM and PIP motifs are important for the Pif1-dependent inhibition of telomerase at DSBs, with the *pif1-rbm* mutation conferring a stronger phenotype than that of *pif1-pip*. This is consistent with the RPA-dependent recruitment of Pif1 to DSBs playing a more important role than the PCNA-dependent mechanism.

In the experiment described above (Fig. 2C), the DSBs were induced in unsynchronous cell populations where the majority of the cells were expected to be in G1 phase. One way to explain the mild effect of *pif1-pip* on telomerase inhibition at DSB could be if PCNA recruited Pif1 to DSBs non-specifically during replication fork passage through broken DNA ends generated in G1. To investigate deeper into this possibility, we synchronised cells in either G1 or G2 prior to the DSB induction. As shown in Fig. 2E, the additional effect of the *pip* mutation on the DNTA can be seen in the absence of the functional RBM motif. This suggests that the PCNA-dependent recruitment of Pif1 to DSB is not coupled to the S-phase genome replication.

The functional role of protein recruitment is often in allowing to overcome a low abundance of a protein by creating a higher local concentration of the protein at the site of its desired activity. Therefore, increasing this concentration by other means, such as overexpression of the gene encoding the recruited protein might compensate for compromised recruitment. Overexpression of the mutant *pif1-m1-rbm* or *pif1-m1-rbm-pip* alleles, which produced Pif1-rbm and Pif1-rbm-pip respectively, using the *GAL1/10* promoter noticeably reduced the frequency of the G418$^S$ Ura$^-$ colonies in the DNTA assay indicating at least a partial suppression of de novo telomeres added to DSBs (Fig. 2C). Therefore, the Pif1 interactions with RPA and PCNA *per se* might be not necessary for the ability of Pif1 to inhibit telomerase. Instead, they may play a role in promoting either Pif1 loading onto ssDNA at the break or its interaction with another protein there, for example, telomerase (Eugster et al, 2006).

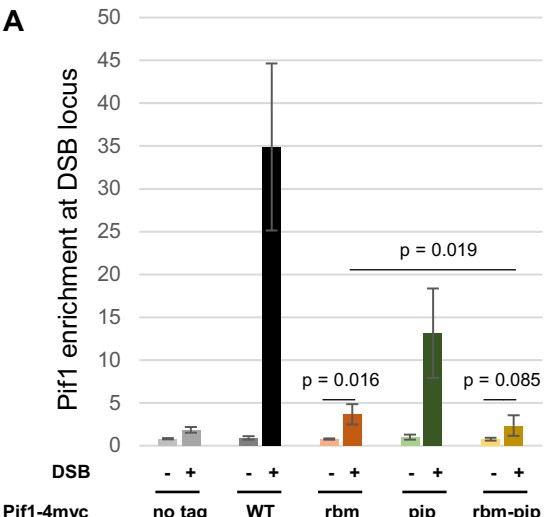

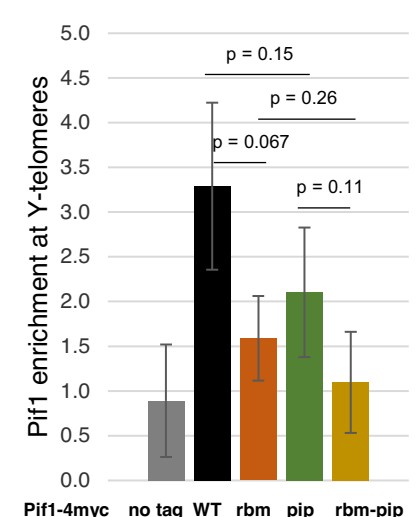

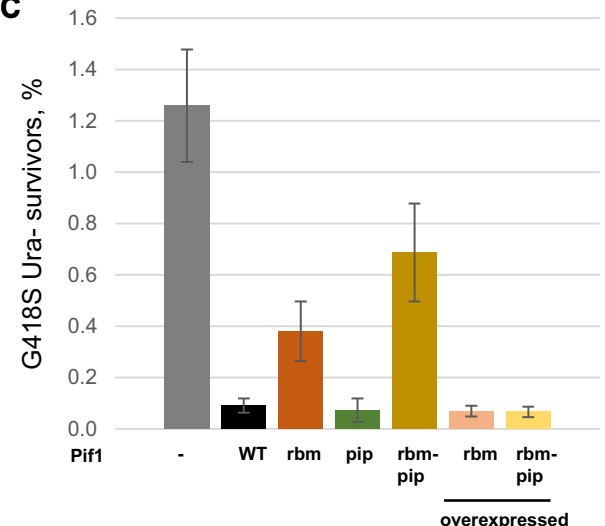

**D**

| Pif1 | DNTA/repair events analysed |
|------|------------------------------|
| - | 11/11 |
| WT | 0/38 |
| rbm | 27/35 |
| pip | 10/56 |

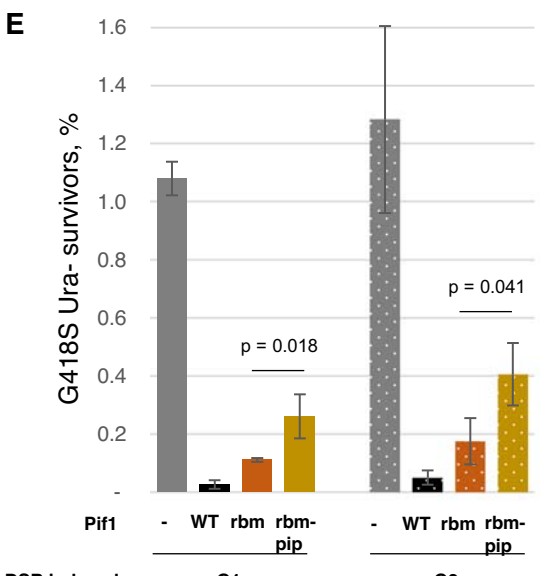

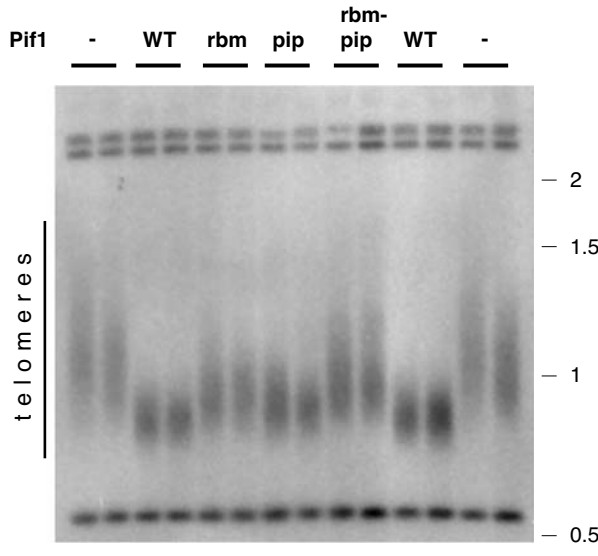

**Figure 2. Both RBM and PIP motifs of Pif1 are required for the Pif1-dependent telomerase inhibition.**

(A) Efficient recruitment of Pif1 to DSBs is dependent on both RBM and PIP motifs. Localisation of Pif1-4myc and its mutant derivatives to the HO-induced DSBs analysed by ChIP-qPCR. The enrichment of Pif1 at *HEM13* (DSBs in YPGAL and no DSBs in YPRAF) relative to the *ARO1* locus (no DSB reference locus used for normalisation) is plotted. Average values $+/-$ SD are plotted ($n = 3$ biological replicates). Statistical significance was calculated by paired *t* test. (B) Pif1 enrichment at chromosome termini requires both RBM and PIP motifs. The enrichment of Pif1 at Y'-telomeres relative to the *ARO1* locus is plotted. Average values $+/-$ SD are plotted ($n = 3$ biological replicates). Statistical significance was calculated by paired *t* test. (C) Pif1-dependent telomerase inhibition at DSB requires the RBM and PIP motifs. The data plot shows the frequency of G418$^S$ Ura$^-$ survivors after the DSB induction. Average values $+/-$ SD are plotted ($n \geq 3$ biological replicates). (D) Randomly picked G418$^S$ Ura$^-$ colonies from the experiment in (C) were analysed by Southern blotting for the presence of de novo telomeres (Fig. EV3B) and the data are summarised in the table. (E) De novo telomere addition in the cells synchronised in either G1 or G2. Average values $+/-$ SD are plotted ($n \geq 3$ biological replicates). Statistical significance was calculated by paired *t* test. (F) Southern blot analysis of the telomere length (teloblot) from the cells expressing different nuclear *PIF1* alleles. The numbers on the right indicate the molecular weight (in kb) of the DNA size marker fragments run alongside the experimental samples. Source data are available online for this figure.

It is worth noticing here that the toxicity of the previously reported Pif1 overexpression (Chang et al, 2009; Nickens et al, 2019) was not due to its helicase activity, but rather due to its recruitment to DNA; this is what enabled us to do the genetic experiments involving Pif1 overexpression. Mutating the RBM motif, but not the catalytic site of Pif1 supressed the toxicity (Fig. EV4). Even the *pip* mutation abolishing the PCNA-dependent recruitment of Pif1 partially rescued the growth defect caused by Pif1 overexpression. Therefore, the mere interaction of Pif1 with its recruiters has a negative effect on cell fitness. This could be due to the excess of Pif1 blocking the docking sites on RPA and PCNA for recruiting vitally important proteins at the sites of DNA replication and possibly DNA repair too.

We next tested if the interactions of Pif1 with RPA and PCNA were also important for the telomerase regulation at telomeres. The telomere length of the *pif1-rbm*, *pif1-pip* and *pif1-rbm-pip* mutants with the equilibrated telomeres was analysed by teloblot, a telomere-specific Southern blotting (Fig. 2F). Consistent with the published data, the telomere length in the cells lacking nuclear Pif1 was noticeably longer than in the cells with the wild-type Pif1 (Schulz and Zakian, 1994). In contrast, the cells with either Pif1-rbm or Pif1-pip had an intermediate telomere length phenotype, suggesting that the Pif1 function at the telomeres was partially compromised in both mutants, but to a greater extent by the loss of the RBM motif than the mutated PIP. The cells expressing Pif1-rbm-pip, a protein with both motifs mutated, had the telomere length similar to the cells lacking nuclear Pif1. These results suggest that the recruitment of Pif1 to telomeres via RPA and PCNA operates as two parallel pathways both of which are important for the efficient telomerase inhibition by Pif1. The phenotypic data addressing the Pif1 function in telomerase inhibition are consistent with the results on the recruitment of Pif1 and the Pif1-dependent telomerase inhibition at DSBs (Fig. 2A–D).

## The RBM and PIP motifs are important for the role of Pif1 in DNA replication

Pif1 has been implicated in promoting DNA replication through a number of hard-to-replicate regions as well as in replication termination (Osmundson et al, 2017; Tran et al, 2017; Chen et al, 2019; Deegan et al, 2019). However, this role of the yeast Pif1 is secondary to that of its homologue Rrm3 and it can be revealed only in the *rrm3Δ* background.

We tested if the PIP and RBM motifs in Pif1 were required for the Pif1-dependent replication fork progression through the highly

expressed tRNA gene *tA(AGC)F*, one of the hard-to-replicate loci, using the *rrm3Δ* strain background and the 2D gel electrophoresis assay adapted from Tran et al, 2017 (Fig. 3A–C). Consistent with the previously published data (Tran et al, 2017), the replication fork pausing at the *tA(AGC)F* locus was increased nearly sixfold in the absence of Rrm3 and over ninefold in the cells lacking both Rrm3 and nuclear Pif1 (Fig. 3B,C). In the *rrm3Δ* background, the cells expressing Pif1-rbm, Pif1-pip or Pif1-rbm-pip mutant proteins all showed similarly increased fork pausing in comparison with the cells containing Pif1-WT.

Next, we asked if the PIP and RBM motifs were required for the role of the Pif1 protein during replication termination in the assay previously described in Deegan et al, 2019 (Fig. 3D,E). X-shaped replication intermediates associated with the defect in replication termination could be observed in the cells lacking Rrm3, and are more pronounced in the double mutants lacking both Rrm3 and Pif1. Similar to the results described above, both RBM and PIP motifs in Pif1 were essential for the ability of the helicase to promote replication termination, as a loss of either one of them fully compromised the Pif1 function at the replication termination sites.

## The RBM and PIP motifs in Pif1 are required for the Pif1 function in BIR

Previous studies have shown that the interaction between Pif1 and PCNA is important during DSB repair by BIR as the *pif1-pip* mutation leads to a reduction in the frequency of BIR characteristic of a complete loss of Pif1 (Buzovetsky et al, 2017). Similar to the *pif1-pip* mutants, cells with Pif1-rbm were also deficient in BIR (Fig. 3F, see Fig. EV3A for the assay schematic). This could be a consequence of the severely compromised recruitment of the Pif1-rbm protein to the BIR sites, similar to the recruitment defect of Pif1-rbm to DSBs, as described above. The cells with the Pif1-rbm-pip derivative showed the BIR deficiency similar to that of the single *rbm* and *pip* mutants, as well as to the yeast with a complete loss of nuclear Pif1. Therefore, the genetic requirements for the Pif1 interactions with RPA and PCNA during BIR resemble those at conventional replication forks and differ from the ones in telomerase inhibition. In all the replication-related functions analysed here, a loss of either RPA or PCNA interaction resulted in the phenotype characteristic of a *pif1* nuclear null mutant, while the same mutants were only partially impaired in telomerase inhibition at DNA breaks and telomeres. In addition, the overexpression of the mutant *pif1-m1-rbm* and *pif1-m1-rbm-pip* alleles failed to rescue the low frequency of BIR in the *pif1-m2*

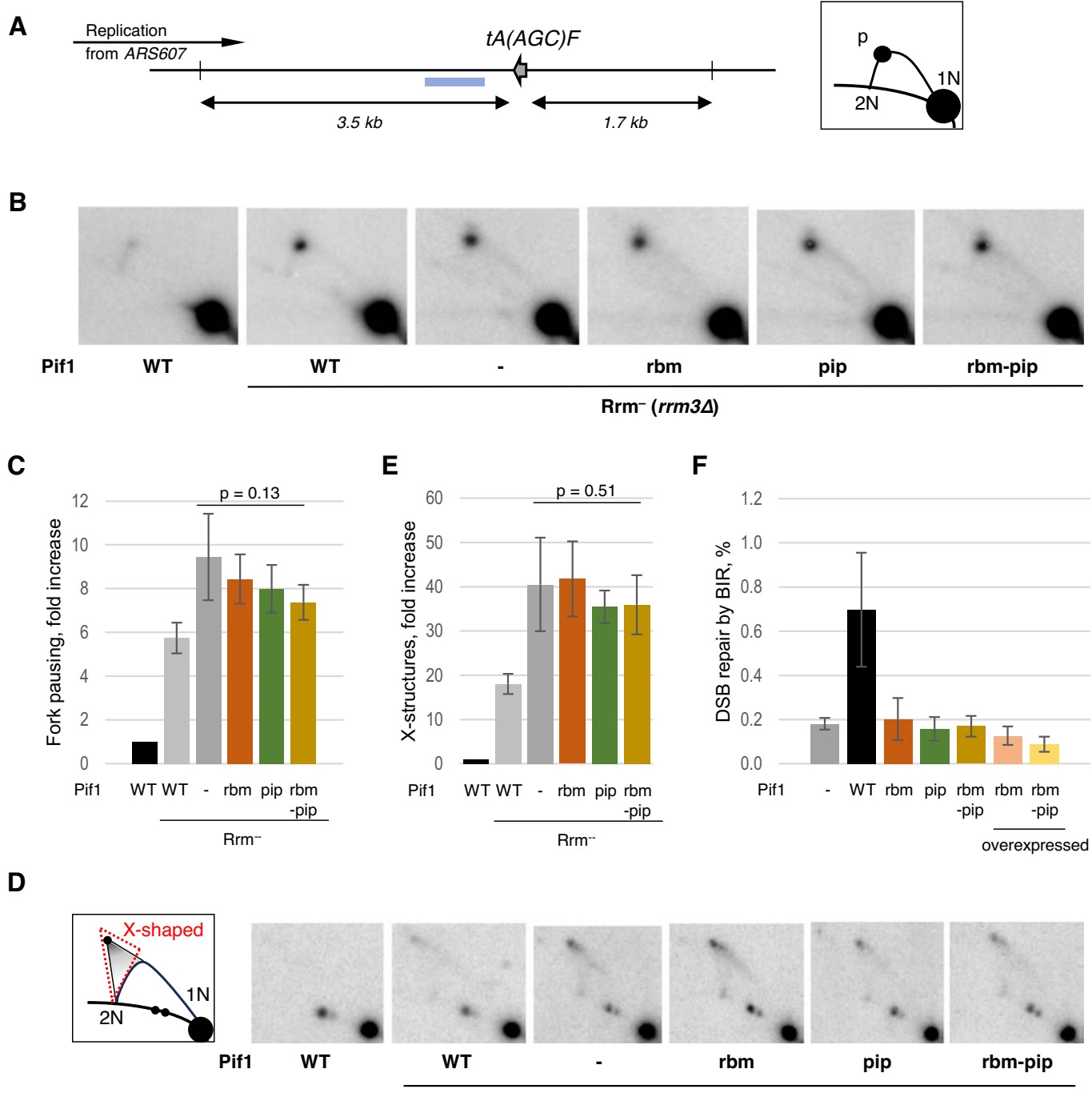

strains (Fig. 3F). These data suggest that the replication-dependent functions of Pif1 are strongly dependent on the interactions between Pif1 and its recruiters to DNA, RPA and PCNA, and might have a functional component where the role of these interactions is beyond creating a higher local concentration of Pif1 at the site of action. The action itself might require Pif1 being bound to both PCNA and RPA.

In this study, we have identified and characterised the Pif1-RPA interaction which is important for a number of the Pif1 functions in the nucleus linked to DNA replication, repair and telomere

maintenance. The role of this interaction parallels the previously reported Pif1-PCNA association (Buzovetsky et al, 2017). Both RPA and PCNA recruit Pif1 to DNA ends. This might be required because of the relatively low abundance of Pif1 characteristic of many helicases and nucleases operating on DNA (Ho et al, 2018). Overexpression of the PIF1 alleles with the mutations in RBM and PIP sequences can functionally suppress the loss of this recruitment, as judged by the telomerase inhibition at DNA breaks (Fig. 2C). This might be explained by another interacting partner for Pif1 on DNA, either a recruiter or telomerase itself, assuming

**Figure 3. The interactions of Pif1 with both RPA and PCNA are essential for the replication-related functions of the Pif1 helicase.**

(A) On the left—a schematic of the genomic region analysed for the replication fork pausing at the *tA(AGC)F* gene locus using 2D gel electrophoresis, as shown in (B, C). The grey arrow depicts the direction of the *tA(AGC)F* transcription. This region is replicated predominantly from the highly active *ARS607* in the direction opposite to the *tA(AGC)F* transcription. The vertical lines indicate the restriction sites for BglII used to digest genomic DNA. The distances between the BglII sites and *tA(AGC)F* are shown by the double-headed arrows. The blue bar shows the probe used to visualise the relevant BglII DNA fragments during Southern blotting. On the right—a schematic of the migration of the replication intermediates within the analysed BglII DNA fragments. Most of the relevant DNA fragments are linear and run at the 1N position. The branched molecules run along the Y-arc projecting from the 1N to the 2N position. The fork pausing at *tA(AGC)F* indicated by P is on the left-hand side of the arc, closer to the 2N spot, because the replication forks moving from the left to the right, as shown on the other schematic, have more than half of the fragment length replicated by the time they are paused at *tA(AGC)F*. (B) Representative Southern blot images of the replication intermediated at the *tA(AGC)F* locus analysed by 2D gel electrophoresis. The Pif1 and Rrm3 status of the used strains is shown under the corresponding images. (C) Quantitative analysis of the replication fork pausing at *tA(AGC)F*. The signal at the pausing site (P) was normalised to the Y-arc signal and then to the corresponding value of the similar ratio in the wild-type strain (the first image on the left in (B)). Therefore, the values on the Y axis show the fold increase in the accumulation of the replication forks at *tA(AGC)F* in different mutants relative to the *RRM3 PIF1* yeast. Averages $+/-$ SD are plotted ($n \geq 3$ biological replicates). Statistical significance was calculated by paired $t$ test. (D) Analysis of the pRS426 plasmid replication by 2D gel electrophoresis. On the left—a schematic of the migration of the pRS426 replication intermediates when linearised with SnaBI which cleaves the circular DNA at the unique restriction site located within the replication origin. The X-shaped structures accumulating due to the replication termination problems run on the 2D gels within the red-dotted triangle. On the right—representative Southern blot images of the replication intermediates of pRS426 analysed by 2D gel electrophoresis. The Pif1 and Rrm3 status of the used strains is shown under the corresponding images. (E) Quantitative analysis of the replication termination intermediates detected by Southern blotting and shown in (D). The probe signal for the X-shaped intermediates (within the red-dotted triangle) was normalised to the signal for the linear DNA fragments (1N) and then to the corresponding value of the wild-type strain (the first image on the left). Averages $+/-$ SD are plotted ($n = 3$ biological replicates). Statistical significance was calculated by paired $t$ test. The probe used for the analysis hybridises to *URA3* present in pRS426. (F) The RBM motif in Pif1 is required for BIR. BIR frequency shown as a percentage of G418$^R$ Ura$^-$ survivor colonies formed after the DSB induction relative to the cell titre in the strains indicated. Average values $+/-$ SD are plotted ($n \geq 3$ biological replicates). Source data are available online for this figure.

the mutated RBM and PIP motifs have no residual low-affinity interactions with the recruiters. Telomerase itself is a strong candidate for being such a partner as the Pif1 interaction with telomerase has been identified (Eugster et al, 2006). Alternatively, the RPA- and PCNA-dependent recruitment may promote Pif1 loading onto ssDNA ends and the subsequent Pif1 movement towards the 3' end might be sufficient for the telomerase inhibition (Fig. 4A). Increasing the concentration of Pif1 through its gene overexpression could promote the loading by simply increasing the number of Pif1 molecules in the vicinity of the DNA.

Interestingly, the role of the RPA and PCNA interactions with Pif1 might be different at replication forks, both conventional and the ones originated by the BIR repair pathway, as both interactions were essential for any genetically detectable Pif1 function there. Pif1 may require a more stable association with the replication machinery at the forks. This is particularly easy to envision for the BIR forks which are required to move efficiently over rather long distances. Any fork stalling may cause a premature ejection of the invaded end, which is constantly being ejected behind the moving fork, resulting in the need of re-assembly of the recombination machinery to re-invade the donor DNA. In this case, RPA and PCNA may play a bigger role than just recruiting Pif1. They might keep Pif1 anchored over the distance of the fork movement (Fig. 4B).

A recently suggested model explaining the RPA-dependent stimulation of the Pif1 helicase activity *in vitro* proposes RPA complexes being pushed along the ssDNA by Pif1 (Mustafi et al, 2023; Mersch et al, 2023) (Fig. 4B). This model would allow Pif1 being bound to the same RPA trimer and move along ssDNA as a single Pif1-RPA complex connected to the moving PCNA through the C-terminus of Pif1. The double anchoring, when the N-terminus of Pif1 is bound to RPA and the C-terminus interacts with PCNA, would greatly decrease the probability of Pif1 dissociating from the fork. In addition, the helicase domain is in a constant contact with the ssDNA and this interaction might be an additional point of contact helping to retain Pif1 at the fork. Such multiple points of interaction at ssDNA have been previously

reported for the BTR/DNA2 complex in human cells, where BTR has several RBMs required for the recruitment to stalled replication forks, while at DSBs another RBM may be provided by DNA2 (Shorrocks et al, 2021). The RPA-binding motifs interact cooperatively with a stretch of RPA complexes on ssDNA and this mode of recruitment plays a role in sensing the accumulation of ssDNA and the recruitment of BTR as a result.

Since the ability of Pif1 to interact with RPA and PCNA affects telomere length, it is tempting to speculate that Pif1 might arrive to telomeres with replication forks. Replication forks are known to stall in the subtelomeric regions and at the telomeres in yeast (Makovets et al, 2004) and this may promote the Pif1 recruitment to the forks at telomeres. This is consistent with the role of Pif1 in promoting the fork movement through G-rich DNA (Paeschke et al, 2011; Ribeyre et al, 2009). Alternatively, Pif1 may be recruited to the chromosome end directly, similar to how it was observed at the induced DSB. Both recruitment pathways may co-exist at telomeres independently.

At the breaks, DNTA becomes Pif1 independent in the *sgs1Δ exo1Δ* double mutant cells lacking the long-range resection (Lydeard et al, 2010; Chung et al, 2010). Such ends have limited amount of RPA and perhaps greatly decreased PCNA, as the PCNA loading relies on the RFC-RPA interaction (Yuzhakov et al, 1999). The low levels of RPA and PCNA might be insufficient to provide Pif1 loading onto the break and this makes Pif1 irrelevant at the broken ends in *sgs1Δ exo1Δ*.

Our findings on the suppression of the growth defect in the cells overexpressing Pif1 through the loss of its RBM and PIP functions, but not the catalytic activity, shed light on the importance of maintaining the low abundance of the DNA metabolism enzymes in the cell. Their presence at the DNA might be regulated through the interactions with the recruiting modules such as PCNA, RPA and Ctf4. Both the N- and the C-termini of Pif1 are known to become phosphorylated in response to DNA damage (Makovets and Blackburn, 2009; Vasianovich et al, 2014; Rossi et al, 2015), and this may regulate the Pif1 recruitment to different loci in the genome. At the same time, increasing

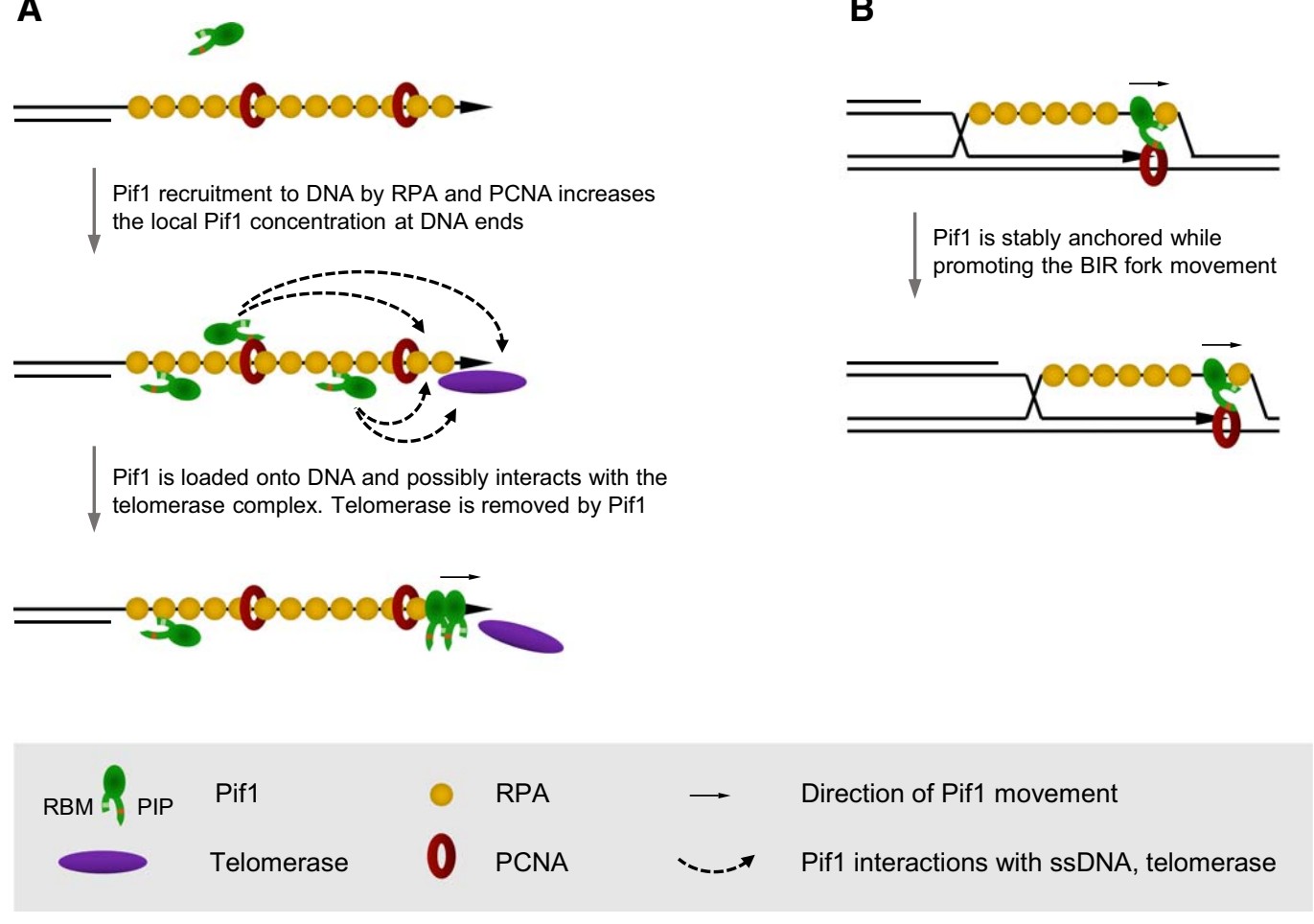

**Figure 4. The functional roles of the Pif1 interactions with RPA and PCNA: molecular models.**

(A) The interactions with RPA and PCNA increase the local concentration of Pif1 at DNA ends, thereby promoting the Pif1-dependent inhibition of telomerase. A resected DSB is shown as an example. At telomeres, Pif1 might be recruited by PCNA and RPA at the replication forks stalled in subtelomeric and telomeric regions. (B) Simultaneous interactions of Pif1 with RPA and PCNA provide a stable association of Pif1 at replication forks. A BIR fork is shown as an example.

the abundance of any of the recruited proteins through gene overexpression might upset the balance of the protein recruitment at the sites of replication and repair by blocking the access of other enzymes to the DNA. Perhaps, this is the reason for the Pif1 toxicity rather than its helicase activity, as suggested by our experiments.

The Pif1 homologue Rrm3 is believed to travel with the replication forks (Azvolinsky et al, 2006), while Pif1 might be recruited specifically to the sites of fork stalling and broken DNA ends. Accumulation of ssDNA bound by RPA at stalled forks and resected breaks might promote the Pif1 recruitment by providing an increasing number of the available docking sites on the N-terminal domain of Rfa1. This mechanism is different from the one providing the recruitment of another site-specific helicase Srs2, which relies on PCNA SUMOylation (Armstrong et al, 2012). Multiple structural changes to both DNA and protein composition might occur at stalled forks. They are then used to recruit additional machinery to restore replication. Understanding the dynamics of different protein complexes at the forks and DNA repair sites is important for unveiling the molecular mechanisms

and the regulatory networks responsible for maintaining genome integrity.

## Methods

### Yeast strains, oligonucleotides and plasmids

Yeast strains, plasmids and oligonucleotides used in this study are described in Tables EV1–3.

### Recombinant protein production

The budding yeast Pif1 fragments were produced in *E. coli* BL21 cells as the C-terminal fusions to GST using pGEX-4T-2 vector (Smith and Johnson, 1988). Rfa1N-4myc was produced in *E. coli* BL21 cells using the pETM-33 vector (Dummler et al, 2005). A single *E. coli* colony was inoculated into the 5 ml LB with ampicillin for pGEX-4T-2 derivatives or kanamycin for the pETM-33-based plasmids and grown overnight at 37 °C. The next day, the cultures

were diluted 50-fold in fresh media and grown at 37 °C with vigorous aeration until mid-log phase ($OD_{600} = 0.5$). The gene expression was induced by the addition of IPTG at 0.1 mM final concentration, and the cells were grown for an additional 3 h, harvested and stored at −80 °C as wet pellets.

## GST pulldowns and co-immunoprecipitation

*E. coli* cell pellets, either freshly obtained or from the −80 °C storage, were resuspended in the cold *E. coli* lysis buffer ELB (50 mM Tris-HCl pH 7.5, 50 mM NaCl, 5 mM EDTA, 1.5 mM BME and cOmplete™ EDTA-free Protease Inhibitor Cocktail, Roche 11836170001) and lysed by sonication using Fisherbrand™ Model 50 Sonic Dismembrator (Fisher scientific, 12961151). The samples were kept on ice between the sonication sessions. After lysis, the cell debris and insoluble material were pelleted by centrifugation at 20,000× g for 10 min at 4 °C, and the cleared lysates were incubated with Glutathion-sepharose 4B beads (Merck, GE17-0756-01) for 2 h at 4 °C on a nutator, to allow the baits to bind the beads. The beads were washed twice with ELB and mixed with either yeast cleared lysates or *E. coli* cleared lysates, containing the preys.

The yeast cells were grown in YPD broth until mid-log phase, pelleted and stored at −80 °C until required. The pellets were resuspended in the yeast lysis buffer (20 mM HEPES pH 7.5, 125 mM NaCl, 0.1% IGEPAL, 5 mM EDTA, cOmplete™ EDTA-free Protease Inhibitor Cocktail, Roche 11836170001) and 1 mM PMSF and lysed by bead beating (Makovets and Blackburn, 2009). The cleared lysate was prepared by centrifugation of the total lysate at 20,000× g for 30 min at 4 °C (to pellet the chromatin) and a small aliquot was taken to analyse along the pull-down samples. For pull-down experiments, the yeast or *E. coli* lysates were added to the glutathione beads with immobilised baits, the samples were incubated for 2 h at 4 °C on a nutator, the beads were washed twice with ELB and boiled in the 1.5× Laemmli buffer for 7 min to denature all proteins. For the co-immunoprecipitation experiment, the yeast lysates were incubated with 9E10 (1:500) for 1 h on ice and then mixed with the Protein G Dynabeads (Invitrogen, 10003D). The beads-lysate mixture samples were incubated for 2 h at 4 °C on a nutator. The beads were washed twice with the lysis buffer without the proteinase inhibitors and boiled in the 1.5× Laemmli buffer. Samples were loaded on SDS-PAGE and analysed by western blotting.

## Western blotting

Yeast whole-cell protein samples were prepared using the TCA precipitation method described in O'Rourke and Herskowitz, 1998. The proteins were resolved on the SDS-PAGE gel and transferred either onto PVDF membrane (Immobilon®-FL, 0.45-µm pores, Merck Millipore Ltd.) or nitrocellulose membrane (Protran BA83, 0.2 µm, Whatman). The proteins tagged with the C-terminal 4myc tag were detected using mouse monoclonal 9E10 antibody (Thermo Fisher Scientific 13-2500, 1:1000 dilution). Rfa1 was detected using rabbit polyclonal anti-RPA antibody (Agrisera AS07 214, 1:20,000 dilution). Act1 (the loading control) was detected using mouse monoclonal anti-beta Actin antibodies (Abcam ab8224, 1:50,000 dilution). GST was detected using mouse monoclonal anti-GST antibody (Abcam, 3G10/1B3). The secondary antibodies used for the western blotting were goat anti-mouse IgG (H + L) cross-

adsorbed secondary antibody, Alexa Fluor 680 (Thermo Fisher Scientific A-21057, 1:12500 dilution) and goat anti-rabbit IgG F(c) IRDye800 Conjugated (Rockland 611-132-003, 1:12500 dilution), respectively. The membranes were scanned using Odyssey® CLx fluorescent scanner (LI-COR®) and analysed with the Image Studio™ Lite software.

## De novo telomere addition and break-induced replication assays

The assays were performed as described previously (Makovets and Blackburn, 2009; Vasianovich et al, 2014). Briefly, the yeast strains with $P_{GAL1}$-HO and the HO-endonuclease recognition site at the *MNT2* locus on chromosome VII (Fig. EV3A) were patched overnight on YP plates with 2% raffinose (YPRAF) to de-repress the $P_{GAL1}$ promoter. Next day, the cells were resuspended in YP broth and serial dilutions were plated on YPD and YP + 2% galactose (YPGAL) plates. The colonies formed on YPGAL were replica-plated on SC -Ura and YPD with G418. The colonies were scored, and the frequencies of the BIR events and the DNTA events were calculated as the ratios of the number of G418$^R$ Ura$^-$ and G418$^S$ Ura$^-$ colonies, respectively, to the number of colonies on YPD plates.

For the break induction in the synchronised populations, the cells were patched on YPRAF overnight, resuspended in liquid YP broth with 2% raffinose and grown for two population doublings to restore logarithmic growth. Each culture was split into two, and the cells were synchronised either in G1 with the addition of α-factor (5 µg/ml) or in G2 with the addition of nocodazole (15 µg/ml). Both cultures were further incubated for 2.5 h in a shaking incubator. Following the synchronisation step, the cultures were split into two again and either galactose (DSB induction) or raffinose (no DSB) was added. Following another hour of incubation on a shaker, the cells were washed with YP broth to remove the synchronising agents and the YPGAL cultures were plated on YPGAL agar, while the YPRAF cultures were plated on YPD agar. The frequencies of DNTA were calculated as described above.

## Spotting assays

Yeast strains were pre-grown on YPRAF plates overnight. The next day, cells were resuspended in YP broth at $OD_{600} = 5$, and a series of fivefold dilutions were spotted on YPD and YPGAL plates using a 48-pin device (frogger). The plates were incubated at 30 °C until colonies were formed—32 h for YPD and 44 h for YPGAL. The images of the colonies were taken using Gel Doc XR+ imaging system (BioRad).

## DSB induction and ChIP

The yeast strains with an HO-inducible DSB at the *HEM13* locus on chromosome IV were patched overnight on YPRAF plates to de-repress the $P_{GAL1}$ promoter. Next day, the cultures were diluted in YPRAF broth and incubated at 30 °C until $OD_{600} = 0.6$, at which point half of the culture was fixed for ChIP as described in Strahl-Bolsinger et al, 1997. The rest was diluted to $OD_{600} = 0.3$ with the fresh pre-warmed YPRAF broth. The *HO* expression was induced by the addition of galactose (2% final concentration). After 3 h incubation, the cells were fixed as above. The ChIP experiments were performed as described previously (Makovets and Blackburn, 2009). Pif1-4myc enrichment at *HEM13* was estimated relative to

the *ARO1* locus unaffected by the DSB induction. Pif1-4myc enrichment at Y'-telomeres was analysed relative to the *ARO1* locus in the unchallenged cells.

## Telomere length analysis

To equilibrate telomere length, the cells were passaged on YPD agar for 100–120 generations (5–6 passages) at 30 °C. Purified genomic DNA was digested with KpnI and resolved on 0.85% agarose gels. Southern blotting and hybridisation were performed as described previously (Makovets et al, 2004).

## Southern blotting analysis of de novo telomere addition

To screen for true DNTA events, randomly picked Ura⁻ G418ˢ colonies from the galactose plates were purified, single colonies were patched on YPD agar and genomic DNA was extracted from the freshly grown patches. The *MNT2-ADH4* region of chromosome VII was analysed by Southern blotting as described previously (Makovets and Blackburn, 2009). Briefly, the purified genomic DNA was digested with EcoRV, resolved on 0.85% agarose gels and assayed by Southern blotting using a radioactively labelled MNT2 probe which detects both fragments of *MNT2* generated after the digest (Fig. EV3). The probes used for the Southern blotting experiments were labelled using a random prime labelling kit (Prime-It II, Agilent Technologies, Santa Clara, CA, USA, 300385) and α-32P-dATP (Perkin Elmer, BLU012-H250UC). DNTA events were identified by the smeared bands characteristic of terminal restriction fragments.

## Analysis of replication fork progression through *tA(AGC)F*

Replication intermediates at the *tA(AGC)F* locus were analysed as described in Tran et al, 2017. Briefly, cells were pre-grown overnight on YPD plates and resuspended in YPD broth next day. The cultures were grown at 30 °C until mid-log phase and cell growth was stopped by the addition of sodium azide (0.1% final concentration) and EDTA (40 mM) followed by 5-min incubation at the growth conditions. Cells were harvested for DNA purification according to the protocol developed in the Brewer lab (http://fangman-brewer.genetics.washington.edu/nib-n-grab.html). 2D gel electrophoresis was performed mainly as described before (Friedman and Brewer, 1995), with the following modifications: the first dimension gels were run at ~0.67 V/cm for 45 h in 0.4% agarose gels without ethidium bromide, and the second dimension gels were run at 3 V/cm for 12 h in 0.9–1% agarose gels with ethidium bromide (0.3 μg/ml) at 4 °C. The DNA was transferred from the gels onto Osmonics MagnaGraph membrane (Maine Manufacturing, 1213502) and hybridised with radioactively labelled probes according to the manufacturers' recommendations. Signal quantification was performed using phosphor-storage screens, a Typhoon FLA 7000 IP2 imager (GE Healthcare, Chalfont St Giles, UK) and ImageQuant TL software (GE Healthcare, version 8.1).

## Analysis of replication termination

pRS426 replication intermediates were analysed as described in Deegan et al, 2019. Briefly, cells were patched overnight on SC-Ura plates and resuspended in the YPD broth the next day at OD₆₀₀ = 0.15. The cultures were propagated for one cell division and the cells were synchronised in G1 with α-factor (0.01 μg/ml) for 3 h. After this, α-factor was washed away and cells were resuspended in fresh YPD broth to release them from the G1 arrest. After 40 min, the cell growth was stopped by the addition of sodium azide (0.1% final concentration) and EDTA (40 mM) followed by 5-min incubation at the growth conditions. Cells were harvested for DNA purification and the 2D gel electrophoresis was performed as described above.

## Statistical analyses

All statistical analyses in this study were performed using a Student's two-sample two-tailed $t$ test. Variances were tested for equality using $f$ test.

# Data availability

This study includes no data deposited in external repositories.

# Peer review information

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

## Acknowledgements

The authors are grateful to Sara Buonomo, Marcus Wilson and the Makovets lab for helpful discussions and suggestions, Tadas Andriuskevicius for critical reading of the manuscript, Tetiana Kardash for help with plasmid construction and yeast plating experiments. This research was funded by the Medical Research Council—Senior Non-Clinical Fellowship, grant number MR/R02068X/1 awarded to SM.

## Author contributions

**Oleksii Kotenko**: Conceptualisation; Formal analysis; Investigation; Visualisation; Methodology; Writing—original draft; Writing—review and editing. **Svetlana Makovets**: Conceptualisation; Resources; Data curation; Supervision; Funding acquisition; Methodology; Project administration; Writing—review and editing.

## Disclosure and competing interests statement

The authors declare no competing interests.

# Expanded View Figures

**A**

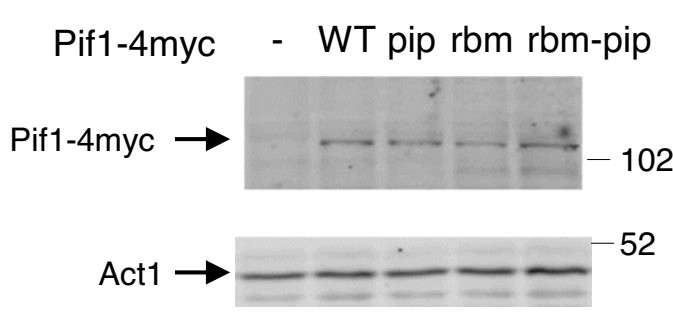

**B**

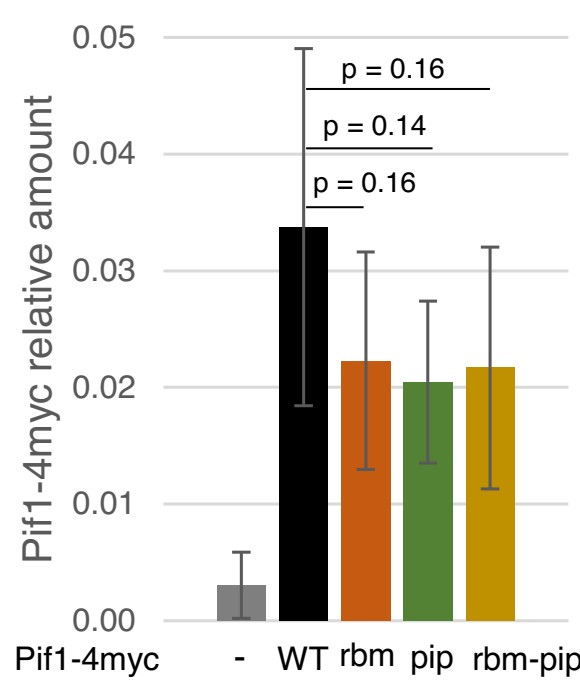

**Figure EV1. Mutations in RBM and PIP motifs of Pif1 do not affect the protein levels.**

(A) Western blot analysis of Pif1 and Act1 from the total cell protein samples prepared from yeast log-phase liquid cultures (a representative image). The numbers on the right indicate the molecular weight (in kDa) of the size marker proteins run alongside the experimental samples. (B) Quantification of the Pif1 steady state levels from the experiments in (A) (normalised to Act1). Average values $+/-$ SD are plotted ($n \geq 3$ biological replicates). Statistical significance was calculated by paired $t$ test. Source data are available online for this figure.

**A**

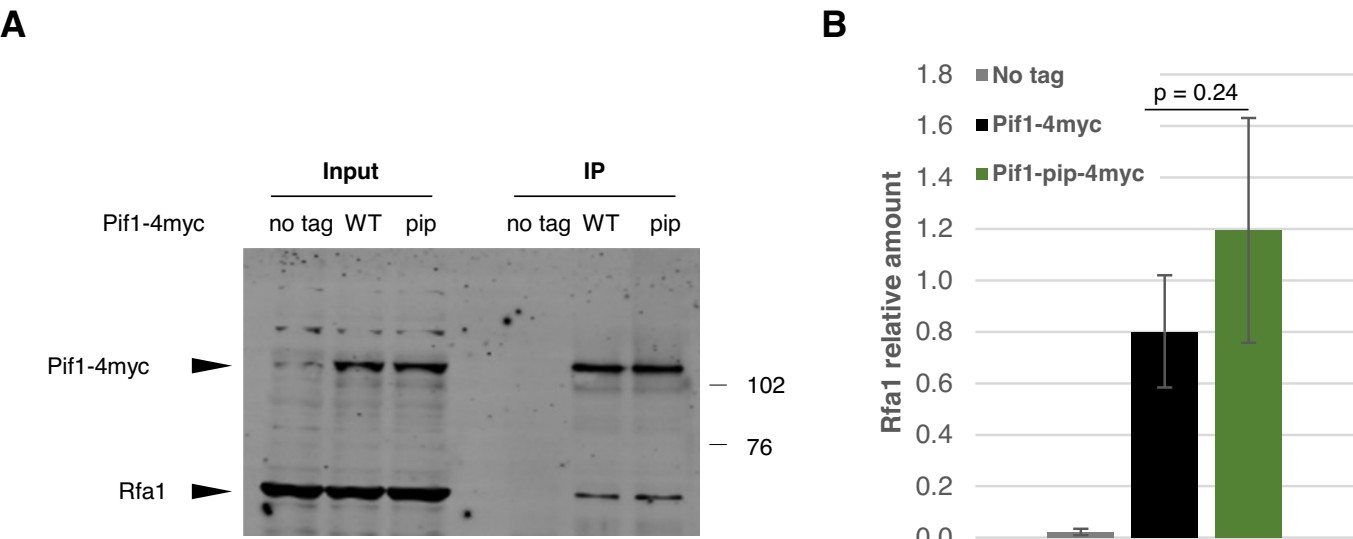

**B**

**Figure EV2.  The PIP motif of Pif1 is not required for the interaction between Pif1 and RPA.**

(A) Proteins immunoprecipitated using anti-myc (9E10) antibodies were analysed by western blotting. Rfa1 was detected using anti-RPA (*S. cerevisiae*) antibody, Pif1-4myc and Pif1-pip-4myc were detected by anti-myc antibody. The numbers on the right indicate the molecular weight of the size marker proteins (in kDa) run alongside the experimental samples. (B) Quantification of the experiments in (A). Rfa1 signal was normalised to the corresponding input and to the relative myc signal. Average values $+/-$ SD are plotted ($n = 3$ biological replicates). Statistical significance was calculated by paired *t* test. Source data are available online for this figure.

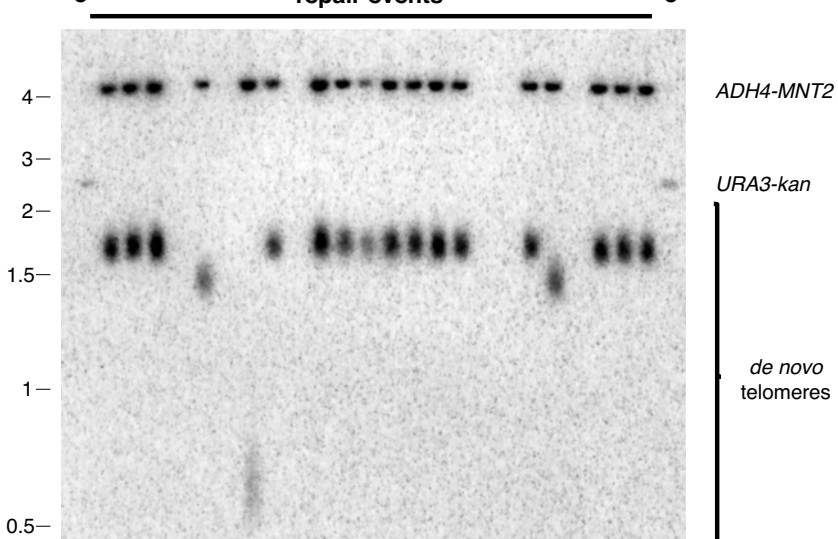

**A**

**Ura⁺ G418ˢ**

HO site

URA3 | kan3' | MNT2 | ADH4 | CEN *CHRVII*

2.4 kb    4.4 kb

*DSB induced*

*Degraded*

URA3 | kan3' | MNT2 | ADH4 | CEN *CHRVII*

*De novo telomere addition* ← → *Homology search*

kan3' | MNT2 | ADH4 | CEN

< 2.4 kb    4.4 kb

**DNTA: Ura⁻ G418ˢ**

0.5 kb

kan3' | MNT2 | ADH4 | CEN *CHRVII*

ARO4 | kan5' | HIS7 | CEN *CHRII*

~ 100 kb

Screening by Southern blotting

*Break-induced replication*

ARO4 | KAN | MNT2 | ADH4 | CEN *CHRVII extended*

~ 100 kb from CHRII

**BIR: Ura⁻ G418ᴿ**

**B**

C    repair events    C

4 —   *ADH4-MNT2*

3 —

2 —   *URA3-kan*

1.5 —

*de novo telomeres*

1 —

0.5 —

◄ **Figure EV3. Analyses of DNTA and BIR.**

(**A**) Schematic of the genetic assay used to analyse the frequency of DSB repair by DNTA and BIR. The DSBs are generated by galactose-inducible expression of the HO-endonuclease in the strains with the HO-recognition site located at *MNT2 (CHRVIIL)*. The modified sub-telomere prior to the DSB induction contains the 3' end of the *KAN* marker (confers a resistance to the drug G418), followed by the HO cleavage site, *URA3* (the endogenous *URA3* is mutated) and a telomere (black rectangle). Prior to the DSB induction, the cells are Ura$^+$ G418$^S$. After the *HO* induction by galactose, the *URA3-telomere* fragment is cleaved off and degraded by the break resection nucleases. The *kan-MNT2* DNA end can be healed by telomerase (the scenario on the left) generating Ura$^-$ G418$^S$ cells, which are then to be screened by Southern blotting (**B**) to detect the actual *de novo* telomeres. For this, the genomic DNA is digested with EcoRV, resolved on an agarose gel and transferred onto a membrane. The DNA is probed with a *MNT2* probe spanning the EcoRV site located within *MNT2*. The lengths of the relevant EcoRV restriction fragments are shown by double arrowheads and the position of the probe on the chromosome by a blue rectangle. Alternatively, the *kan-MNT2* end can be repaired by BIR (the scenario on the right) involving a 0.5 kb *KAN* homology provided by the overlapping sequences between two incomplete versions of *KAN* indicated by the dashed lines. The completion of BIR leads to a reconstitution of the full-length *KAN* and an extension of the *CHRVIIL* arm by ~100 kb, due to copying the DNA sequence of *CHRII* from the point of the break invasion to the telomere. This generates Ura$^-$ G418$^R$ colonies. (**B**) A representative image of a Southern blot used to screen for Ura$^-$ G418$^S$ colonies for DNTA events. Lanes marked with C contain samples from the cells before the break induction. The numbers on the left indicate the molecular weight (in kb) of the DNA size marker fragments run alongside the experimental samples. Source data are available online for this figure.

**A**

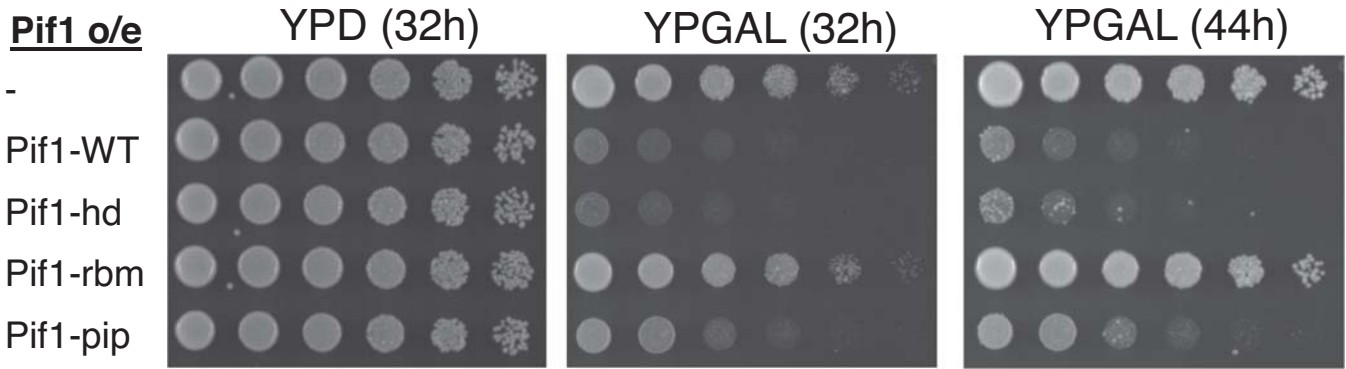

**B**

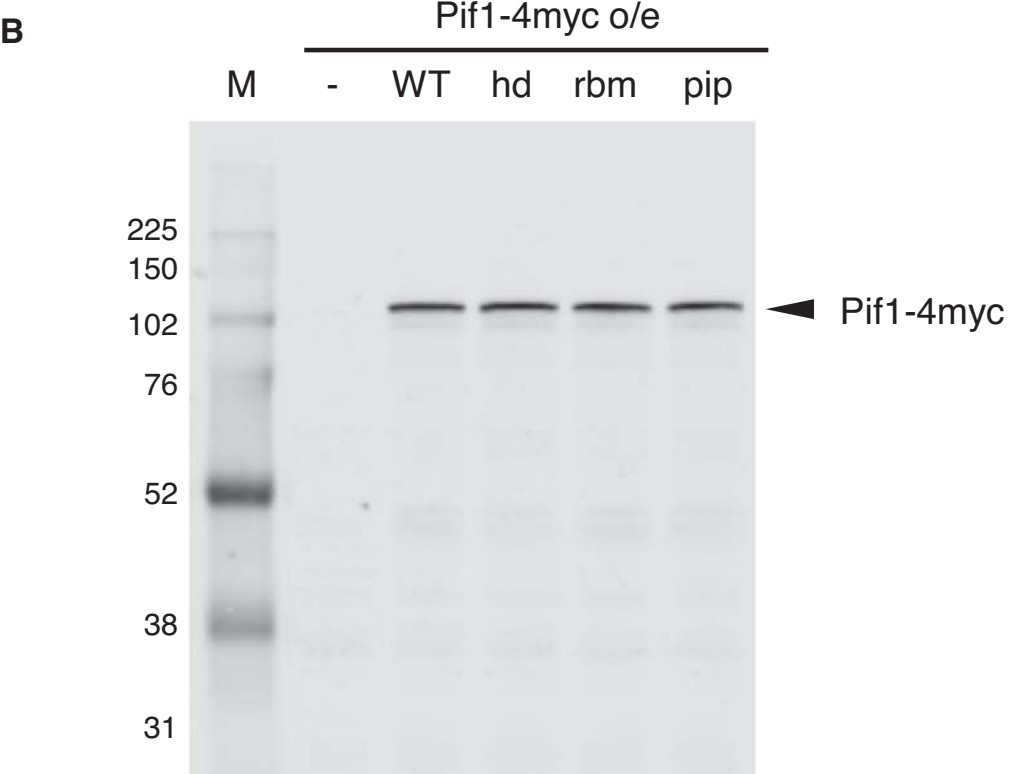

**Figure EV4.** Overexpression of Pif1 is toxic due to its ability to interact with RPA and PCNA. (**A**) Fivefold serial dilutions of the freshly grown isogenic strains were spotted on YPD and YPGAL plates and incubated at 30 °C for the duration shown above the images. (**B**) Comparative analysis of the relative Pif1-4myc protein levels in YPGAL for the strains analysed in panel A. Galactose was added to the log-phase cultures grown in YPRAF and after 3 h of additional culturing cells were harvested and the total cell lysates were analysed by western blotting. Pif1-hd is a helicase-dead Pif1 derivative containing the previously characterised K264A substitution.

