## [Peer Review File · EMBO Reports]

The functional significance of the RPA- and PCNA-dependent recruitment of Pif1 to DNA

Svetlana Makovets and Oleksii Kotenko

Corresponding author(s): Svetlana Makovets (smakovet@staffmail.ed.ac.uk)

Review Timeline:

Submission Date:	19th Aug 23
Editorial Decision:	13th Oct 23
Revision Received:	13th Jan 24
Editorial Decision:	8th Feb 24
Revision Received:	10th Feb 24
Accepted:	26th Feb 24

Editor: Deniz Senyilmaz Tiebe

Transaction Report:

Dear Dr. Makovets,

Thank you for the submission of your research manuscript to our journal, which was now seen by three referees, whose reports are copied below.

My apologies for this unusual delay in getting back to you. It took longer than anticipated to receive the full set of referee reports.

We concur with the referees that the proposed role of RPA in Pif1 recruitment is in principle very interesting. However, they also raise significant concerns that need to be addressed to consider publication here.

Given these positive recommendations, we would like to invite you to submit a revised manuscript. Please revise your manuscript with the understanding that the referee concerns (as in their reports) must be fully addressed and their suggestions taken on board. Please address all referee concerns in a complete point-by-point response. Acceptance of the manuscript will depend on a positive outcome of a second round of review. It is EMBO reports policy to allow a single round of major experimental revision only and acceptance or rejection of the manuscript will therefore depend on the completeness of your responses included in the next, final version of the manuscript.

We realize that it is difficult to revise to a specific deadline. In the interest of protecting the conceptual advance provided by the work, we recommend a revision within 3 months. Please discuss the revision progress ahead of this time with me if you require more time to complete the revisions, or if you have questions or comments regarding the revision (also by video chat).

1. A data availability section providing access to data deposited in public databases is missing (where applicable).
2. Your manuscript contains statistics and error bars based on $n=2$. Please use scatter plots in these cases.

You can submit the revision either as a Scientific Report or as a Research Article. For Scientific Reports, the revised manuscript can contain up to 5 main figures and 5 Expanded View figures, and it should not exceed 27000 characters. If the revision leads to a manuscript with more than 5 main figures it will be published as a Research Article. In this case the Results and Discussion section should be separate. If a Scientific Report is submitted, these sections have to be combined. This will help to shorten the manuscript text by eliminating some redundancy that is inevitable when discussing the same experiments twice. In either case, all materials and methods should be included in the main manuscript file.

4) a .docx formatted letter INCLUDING the reviewers' reports and your detailed point-by-point responses to their comments. As part of the EMBO publication's Transparent Editorial Process, EMBO reports publishes online a Review Process File (RPF) to accompany accepted manuscripts. This File will be published in conjunction with your paper and will include the referee reports, your point-by-point response and all pertinent correspondence relating to the manuscript.

<https://www.embopress.org/page/journal/14693178/authorguide#transparentprocess>

You are able to opt out of this by letting the editorial office know (emboreports@embo.org). If you do opt out, the Review

Process File link will point to the following statement: "No Review Process File is available with this article, as the authors have chosen not to make the review process public in this case."

5) a complete author checklist, which you can download from our author guidelines

<https://www.embopress.org/page/journal/14693178/authorguide>. Please insert information in the checklist that is also reflected in the manuscript. The completed author checklist will also be part of the RPF.

6) Please note that all corresponding authors are required to supply an ORCID ID for their name upon submission of a revised manuscript (. Please find instructions on how to link your ORCID ID to your account in our manuscript tracking system in our Author guidelines

7) Before submitting your revision, primary datasets produced in this study need to be deposited in an appropriate public database (see <https://www.embopress.org/page/journal/14693178/authorguide#datadeposition>). Please remember to provide a reviewer password if the datasets are not yet public. The accession numbers and database should be listed in a formal "Data Availability" section placed after Materials & Method (see also

<https://www.embopress.org/page/journal/14693178/authorguide#datadeposition>). Please note that the Data Availability Section is restricted to new primary data that are part of this study. * Note - All links should resolve to a page where the data can be accessed. *

Additional information on source data and instruction on how to label the files are available:

<https://www.embopress.org/page/journal/14693178/authorguide#sourcedata>

9) Our journal encourages inclusion of *data citations in the reference list* to directly cite datasets that were re-used and obtained from public databases. Data citations in the article text are distinct from normal bibliographical citations and should directly link to the database records from which the data can be accessed. In the main text, data citations are formatted as follows: "Data ref: Smith et al, 2001" or "Data ref: NCBI Sequence Read Archive PRJNA342805, 2017". In the Reference list, data citations must be labeled with "[DATASET]". A data reference must provide the database name, accession number/identifiers and a resolvable link to the landing page from which the data can be accessed at the end of the reference. Further instructions are available at <http://www.embopress.org/page/journal/14693178/authorguide#referencesformat>

10) Regarding data quantification (see Figure Legends:

<https://www.embopress.org/page/journal/14693178/authorguide#figureformat>)

12) Please also note our reference format:

I look forward to seeing a revised version of your manuscript when it is ready. Please let me know if you have questions or comments regarding the revision.

Kind regards,

Deniz Senyilmaz Tiebe

Deniz Senyilmaz Tiebe, PhD
Scientific Editor
EMBO Reports

Referee #1:

Pif1 is a helicase with many cellular functions, including telomerase inhibition, maintenance of mitochondrial DNA, Okazaki fragment maturation, break-induced replication, and unwinding of G-quadruplexes. In this manuscript, the authors identified RPA as a major recruiter of Pif1 to DNA. This recruitment was abolished by mutation of a four-amino acid motif in the N-terminus of Pif1. They determined that recruitment of Pif1 by RPA functions in parallel to the previously identified Pif1-PCNA interaction, which involves the C-terminal region of Pif1. At double-strand breaks and telomeres, RPA and PCNA are both partially required for telomerase inhibition; abolishing both recruitment pathways mimics a complete loss of nuclear Pif1. In contrast, both RPA and PCNA recruitment pathways are essential for Pif1's role in DNA replication (in the absence of Rrm3 at a hard-to-replicate tRNA gene and a plasmid replication termination site, as well as in break-induced replication). Overall, these discoveries advance our understanding about Pif1, a multifaceted, evolutionarily conserved helicase. A few suggestions that could improve the manuscript:

1. The first paragraph of page 7 discusses the RPA- and PCNA-dependent recruitment of Pif1 to telomeres to regulate telomerase and telomere length. Specifically, the authors propose that Pif1 may be recruited to stalled replication forks traversing telomeric repeats. While true, it is also important to keep in mind that telomeres are unique in that Pif1 is likely recruited to telomeres to perform multiple functions, and that recruitment can occur independently via the replication fork and the chromosome end. It is interesting that abrogating the PCNA-Pif1 interaction has an effect on de novo telomere addition (Figure 2A). In their experiment, the HO break presumably occurs in G1 phase (as cells come out of stationary phase). Since telomerase is not active in G1, DNTA would only occur during/after replication proceeds to the HO break-and this may be how PCNA contributes to Pif1-mediated inhibition of DNTA. What would happen if the experiment is performed on G2-arrested cells? In this case, there should be no replication forks so the Pif1 pip mutant should have no effect. Such an experiment would confirm independent recruitment of Pif1 to telomeres via the replication fork and the chromosome end.
2. Also concerning the first paragraph of page 7, it is debatable whether "Pif1 predominantly inhibits telomerase at longer telomeres"-see PMID: 29052759.
3. From looking at the 2D gels shown in Figure 3B, it does not appear that the Pif1 mutants increase fork pausing (as quantified in Figure 3C). It is clear that these are merely representative blots, and I do not question the results and quantification, but it would be nice if the 2D gels and their quantification matched more obviously (like, for example, in Figures 3D and 3E).
4. On page 6, the authors suggest that Pif1 may interact with telomerase itself. Evidence for this has previously been reported and could be cited (PMID: 16878131).

Referee #2:

In the manuscript titles „The functional significance of the RPA- and PCNA-dependent recruitment of Pif1 to DNA“, Kotenko and Makovets demonstrate how the Pif1 helicase is recruited to sites of double strand breaks (DSBs), stressed replication forks, and telomeres. In Figure 1, the authors use biochemical means to identify a motif in the N-terminal disordered region of Pif1 which can bind to RPA. This RBM is both sufficient, and necessary for the interaction as demonstrated by co-precipitation reactions from purified protein and whole cell extracts. Also using purified proteins, this interaction, and the dependency on the RBM was demonstrated, suggesting that the interaction is direct. Using, ChIP assays, the authors could show that Pif1 was able to be cross-linked to DSBs and that both the PCNA and RPA interaction motifs were required, with the RBN playing a more prominent role. The RBN was also more important to prevent de novo telomere addition at DSBs, consistent with the ChIP results. This was also consistent with the telomere length data, i.e. that both domains on Pif1 were important to prevent telomerase-mediated

elongation of telomeres. At natural pause sites there was an equal requirement of the PIP-box and the RBN to suppress replication pausing and the accumulation of X-shaped intermediates, as assayed by 2D gels in the absence of Rrm3. Finally, the authors show that BIR requires both domains equally. Taken together the authors suggest that Pif1 needs to be stably associated to stalled replication forks to promote elongation during BIR, whereas a loose (potentially two step) recruitment of Pif1 is occurring at telomeres and double strand breaks to prevent telomere addition.

In general, this is a nice manuscript and the experiments performed are convincing. I question the discussion a bit. The authors speculate that Pif1 may be kept away from short telomeres because replication stress is alleviated. However, it has been shown by the Cooper and Coulon labs (in *S. pombe*) that replication stress is increased at short telomeres. Moreover, the Luke lab has shown (in *S. cerevisiae*) that RNA-DNA hybrids accumulate at short telomeres, which likely also increased replication stress....based on these data one would expect to see more Pif1 at short telomeres. On the other hand, we know that critically short telomeres do get elongated by BIR to prevent premature replicative senescence. Hence, I think it would be interesting to look at rates of senescence in the *pif1* mutant alleles, when telomerase is absent. Moreover, it would be interesting to look at Pif1 recruitment to telomeres in the absence and presence of telomerase in the context of the PIP and RBM mutants. This may help the authors to make more concrete statements.

Referee #3:

Pif1-family DNA helicases are multi-functional enzymes with critical roles in telomere regulation, break-induced replication (BIR), replication of structured DNA, and DNA replication termination. Despite their key role in numerous genome maintenance pathways, very little is known about the molecular mechanisms by which Pif1-family helicases are recruited to chromosomes or the DNA replication machinery. In this manuscript, the authors present compelling evidence that the N-terminal region of Pif1 contains an RPA-binding motif, which is required for Pif1 function in various pathways. Although this reviewer has some reservations about the data relating to the requirement of RPA and PCNA-binding in DNA replication at tRNA genes and replication termination (Figure 3, expanded on below), the other data are convincing, and represent a strong contribution to the genome stability field. Overall, this manuscript will be a strong candidate for publication in EMBO Reports, provided the authors can address the reviewer's comments.

Major points:

- The evidence that the RPA and PCNA-binding motifs are required for Pif1 recruitment to double-strand breaks, telomerase inhibition, and BIR is strong. However, the 2D gel analyses (Figure 3), describing the requirement for these interactions in tRNA replication and replication termination are less convincing. To this reviewer's eye, there is little to no difference in the signal of the pausing spot (Fig. 3B) in any of the *rrm3 pif1* double mutants compared to the *rrm3* control strain, especially when considered relative to the signal of the 1N spot. Likewise, the 2D gels for the Pif1 mutants (-, *rmb*, *pip*, *rbm-pip*) in Fig. 3D all look to have stronger overall signal than the *rrm3* control, rather than any specific accumulation of X-shaped molecules. Even if one believes the quantifications presented in 3C and 3E, the accumulation of pausing / termination intermediates caused by Pif1 disruption is fairly minor (~1.3-1.5-fold in 3C, ~2-fold in 3E). At a very minimum, the authors should provide multiple exposures of each 2D gel, to allow the reviewers (and readers) to properly assess the differences between control and mutant strains. If the impact of RPA-binding and PCNA-binding mutations on tRNA replication and termination (Figure 3) is mild compared to the impacts of telomerase displacement and BIR, the authors might consider the possibility that alternative recruitment mechanisms exist for replisome-dependent (e.g. tRNA replication, replication termination) and -independent (e.g. telomerase displacement, BIR) functions of Pif1. This would inform the model presented in Figure 4.
- Related to the above point, the quantification of the 2D gel experiments should be performed in the same way in Figure 3C and 3E. Figure 3C is quantified relative to a strain that is wildtype for both Rrm3 and Pif1, whereas Figure 3E is quantified relative to an *rrm3* strain. This is confusing for the reader. Related to this point, Figure 3D needs the control experiment of a strain that is wildtype for both Rrm3 and Pif1
- A control experiment to test if the various mutations introduced into Pif1 impact Pif1 function non-specifically would be helpful. The authors could purify Pif1 (and the RPA and PCNA-binding mutants) and test helicase activity in vitro. Alternatively, it would be helpful if the authors test the capacity of Pif1 RPA-binding mutants to bind PCNA, and vice versa.
- Figure 1A needs a loading control (e.g. anti-GST western blot)

Minor points:

- Is the RPA-binding motif evolutionarily conserved in Pif1-family helicases from other eukaryotes?

Authors' responses to Reviewers' comments to Kotenko and Makovets, 2024 (EMBOR-2023-58018V1)

Referee #1:

Pif1 is a helicase with many cellular functions, including telomerase inhibition, maintenance of mitochondrial DNA, Okazaki fragment maturation, break-induced replication, and unwinding of G-quadruplexes. In this manuscript, the authors identified RPA as a major recruiter of Pif1 to DNA. This recruitment was abolished by mutation of a four-amino acid motif in the N-terminus of Pif1. They determined that recruitment of Pif1 by RPA functions in parallel to the previously identified Pif1-PCNA interaction, which involves the C-terminal region of Pif1. At double-strand breaks and telomeres, RPA and PCNA are both partially required for telomerase inhibition; abolishing both recruitment pathways mimics a complete loss of nuclear Pif1. In contrast, both RPA and PCNA recruitment pathways are essential for Pif1's role in DNA replication (in the absence of Rrm3 at a hard-to-replicate tRNA gene and a plasmid replication termination site, as well as in break-induced replication). Overall, these discoveries advance our understanding about Pif1, a multifaceted, evolutionarily conserved helicase. A few suggestions that could improve the manuscript:

1. The first paragraph of page 7 discusses the RPA- and PCNA-dependent recruitment of Pif1 to telomeres to regulate telomerase and telomere length. Specifically, the authors propose that Pif1 may be recruited to stalled replication forks traversing telomeric repeats. While true, it is also important to keep in mind that telomeres are unique in that Pif1 is likely recruited to telomeres to perform multiple functions, and that recruitment can occur independently via the replication fork and the chromosome end.

The paragraph mentioned above was modified by addition of the following: "Alternatively, Pif1 may be recruited to the chromosome end directly, similar to how it was observed at the induced DSB. Both recruitment pathways may co-exist at telomeres independently."

It is interesting that abrogating the PCNA-Pif1 interaction has an effect on de novo telomere addition (Figure 2A). In their experiment, the HO break presumably occurs in G1 phase (as cells come out of stationary phase). Since telomerase is not active in G1, DNTA would only occur during/after replication proceeds to the HO break-and this may be how PCNA contributes to Pif1-mediated inhibition of DNTA. What would happen if the experiment is performed on G2-arrested cells? In this case, there should be no replication forks so the Pif1 pip mutant should have no effect. Such an experiment would confirm independent recruitment of Pif1 to telomeres via the replication fork and the chromosome end.

We welcomed this interesting hypothesis and addressed it by doing the proposed experiment, along with a parallel variation with G1 arrested cells (replication fork travels through the break), to compare the two conditions (data added in Figure 2E). The pip-mutation still had an additional effect in the absence of the RBM motif suggesting that PIP was involved in Pif1 recruitment to DSBs independently of replication forks.

2. Also concerning the first paragraph of page 7, it is debatable whether "Pif1 predominantly inhibits telomerase at longer telomeres"-see PMID: 29052759.

The debatable statement "Because the fork stalling is alleviated at short telomeres, Pif1 is less likely to localise to shorter telomeres. This could explain why Pif1 predominantly inhibits telomerase at longer telomeres (Phillips et al., 2015)" has been removed.

3. From looking at the 2D gels shown in Figure 3B, it does not appear that the Pif1 mutants increase fork pausing (as quantified in Figure 3C). It is clear that these are merely representative blots, and I do not question the results and quantification, but it would be nice if the 2D gels and their quantification matched more obviously (like, for example, in Figures 3D and 3E).

Indeed, the visual perception in these experiments depends on how evenly the samples are loaded. We replaced the images with other ones where the loading was more even (Figure 3B).

4. On page 6, the authors suggest that Pif1 may interact with telomerase itself. Evidence for this has previously been reported and could be cited (PMID: 16878131).

We are grateful for this suggestion. The relevant text was modified and the brought up reference was included in the manuscript as suggested.

Referee #2:

In the manuscript titles „The functional significance of the RPA- and PCNA-dependent recruitment of Pif1 to DNA", Kotenko and Makovets demonstrate how the Pif1 helicase is recruited to sites of double strand breaks (DSBs), stressed replication forks, and telomeres. In Figure 1, the authors use biochemical means to identify a motif in the N-terminal disordered region of Pif1 which can bind to RPA. This RBM is both sufficient, and necessary for the interaction as demonstrated by co-precipitation reactions from purified protein and whole cell extracts. Also using purified proteins, this interaction, and the dependency on the RBM was demonstrated, suggesting that the interaction is direct. Using, ChIP assays, the authors could show that Pif1 was able to be cross-linked to DSBs and that both the PCNA and RPA interaction motifs were required, with the RBM playing a more prominent role. The RBM was also more important to prevent de novo telomere addition at DSBs, consistent with the ChIP results. This was also consistent with the telomere length data, i.e. that both domains on Pif1 were important to prevent telomerase-mediated elongation of telomeres. At natural pause sites there was an equal requirement of the PIP-box and the RBM to suppress replication pausing and the accumulation of X-shaped intermediates, as assayed by 2D gels in the absence of Rrm3. Finally, the authors show that BIR requires both domains equally. Taken together the authors suggest that Pif1 needs to be stably associated to stalled replication forks to promote elongation during BIR, whereas a loose (potentially two step) recruitment of Pif1 is occurring at telomeres and double strand breaks to prevent telomere addition.

In general, this is a nice manuscript and the experiments performed are convincing. I question the discussion a bit. The authors speculate that Pif1 may be kept away from short telomeres because replication stress is alleviated. However, it has been shown by the Cooper and Coulon labs (in *S. pombe*) that replication stress is increased at short telomeres. Moreover, the Luke lab has shown (in *S. cerevisiae*) that RNA-DNA hybrids accumulate at short telomeres, which likely also increased replication stress....based on these data one would expect to see more Pif1 at short telomeres.

We are grateful for this remark which is similar to the one by another reviewer. Following the feedback from Reviewers 1 and 2, we have removed the debatable statement from the text.

On the other hand, we know that critically short telomeres do get elongated by BIR to prevent premature replicative senescence. Hence, I think it would be interesting to look at rates of senescence in the *pif1* mutant alleles, when telomerase is absent.

We felt that this experiment wouldn't add much to the direct assay of BIR presented in Figure 3F and the established knowledge that Pif1 is required for type I survivor formation. Since Pif1 is not required for type II survivors, the accelerated senescence observed in *rad52* mutants wouldn't be as strongly affected by the *pif1* mutations.

Moreover, it would be interesting to look at Pif1 recruitment to telomeres in the absence and presence of telomerase in the context of the PIP and RBM mutants. This may help the authors to make more concrete statements.

We performed the proposed experiments and added the data. The effect of the *rbm* and *pip* mutations on the Pif1 recruitment to the Y'-telomeres was addressed by ChIP-qPCR and added to Figure 2 (see Figure 2B).

Given that Pif1 interacts with the telomerase components (PMID: 16878131; Kotenko and Makovets unpublished), we believe that performing this experiment in the cells lacking telomerase is a very intriguing question worth of a separate investigation, but peripheral to the main message of this paper.

Referee #3:

Pif1-family DNA helicases are multi-functional enzymes with critical roles in telomere regulation, break-induced replication (BIR), replication of structured DNA, and DNA replication termination. Despite their key role in numerous genome maintenance pathways, very little is known about the molecular mechanisms by which Pif1-family helicases are recruited to chromosomes or the DNA replication machinery. In this manuscript, the authors present compelling evidence that the N-terminal region of Pif1 contains an RPA-binding motif, which is required for Pif1 function in various pathways. Although this reviewer has some reservations about the data relating to the requirement of RPA and PCNA-binding in DNA replication at tRNA genes and replication termination (Figure 3, expanded on below), the other data are convincing, and represent a strong contribution to the genome stability field. Overall, this manuscript will be a strong candidate for publication in EMBO Reports, provided the authors can address the reviewer's comments.

Major points:

- The evidence that the RPA and PCNA-binding motifs are required for Pif1 recruitment to double-strand breaks, telomerase inhibition, and BIR is strong. However, the 2D gel analyses (Figure 3), describing the requirement for these interactions in tRNA replication and replication termination are less convincing. To this reviewer's eye, there is little to no difference in the signal of the pausing spot (Fig. 3B) in any of the *rrm3Δ pif1* double mutants compared to the *rrm3Δ* control strain, especially when considered relative to the signal of the 1N spot.

We agree with the Reviewer's criticism that representative images in Figure 3B are hard to interpret by eye, because the differences between the relevant samples are within 1.5-fold for Pif1-WT and Pif1-nuclear null in *rrm3* mutant background. To help the readers, we replaced the images with the ones with more even loading.

On a separate note, it is important to mention that the normalisation to the 1N spot cannot be used in this set of experiments. The *rrm3* cells have a different FACS profile compared to the *Rrm3*⁺ cells which suggest that the cells undergo a transient arrest after the bulk of DNA replication is finished, probably due to the hard-to-replicate regions that require *Rrm3* for the efficient replication (see PMID: 30442759). This phenotype is further exacerbated in the *rrm3 pif1-m2* double mutant and additionally manifested in the cell growth defect. Because of these arrests, the *rrm3* and *rrm3 pif1-m2* cells replicate less frequently (i.e. longer doubling time) and therefore the replication forks, paused or not, are less frequent in this populations. If the pausing is normalised to 1N, i.e. non-replicating DNA, it will lead to underestimated values in these mutants. To overcome this problem, we chose to use the signal in the Y-arc, i.e. non-pausing forks, for normalization across the strains as it gives a better account of the active replication forks passing through the locus. This is in agreement with the approach which was used in the original publication introducing this method (PMID: 28429714).

Likewise, the 2D gels for the Pif1 mutants (-, *rmb*, *pip*, *rbm-pip*) in Fig. 3D all look to have stronger overall signal than the *rrm3Δ* control, rather than any specific accumulation of X-shaped molecules. Even if one believes the quantifications presented in 3C and 3E, the accumulation of pausing / termination intermediates caused by Pif1 disruption is fairly minor (~1.3-1.5-fold in 3C, ~2-fold in 3E). At a very minimum, the authors should provide multiple exposures of each 2D gel, to allow the reviewers (and readers) to properly assess the differences between control and mutant strains.

We agree with the Reviewer's criticism. The 2D gel experiments in Figure 3D were repeated with the attempt to achieve a more even loading and these were used to replace the images in the original version. We also provided full uncropped images containing additional repeats with lower DNA loading (see source data for Figure 3D)

If the impact of RPA-binding and PCNA-binding mutations on tRNA replication and termination (Figure 3) is mild compared to the impacts of telomerase displacement and BIR, the authors might consider the possibility that alternative recruitment mechanisms exist for replisome-dependent (e.g. tRNA replication, replication termination) and -independent (e.g. telomerase displacement, BIR) functions of Pif1. This would inform the model presented in Figure 4.

The might have been a mis-interpretation of our data by the reviewer because of how the data plots look. In fact, the effect of the *rbm* and *pip* mutations on the tRNA replication and termination mimics that of a nuclear *pif1* null (*pif1-m2*) and is indistinguishable statistically from *pif1-m2*. We added the p-values to the charts (Figure 3C and E) to avoid the confusion.

- Related to the above point, the quantification of the 2D gel experiments should be performed in the same way in Figure 3C and 3E. Figure 3C is quantified relative to a strain that is wildtype for both Rrm3 and Pif1, where as Figure 3E is quantified relative to an *rrm3Δ* strain. This is confusing for the reader. Related to this point, Figure 3D needs the control experiment of a strain that is wildtype for both Rrm3 and Pif1.

We agree with the Reviewer's suggestion. The experiment was repeated with the WT control samples added in multiple repeats, the representative images now include an image for this control and the quantification analysis was re-done so that all the calculations are relative to this WT control (see Figure 3D-E).

- A control experiment to test if the various mutations introduced into Pif1 impact Pif1 function non-specifically would be helpful. The authors could purify Pif1 (and the RPA and PCNA-binding mutants) and test helicase activity *in vitro*. Alternatively, it would be helpful if the authors test the capacity of Pif1 RPA-binding mutants to bind PCNA, and vice versa.

We appreciate the Reviewer's suggestion. To address if the *pip* mutation affects interaction between the full length Pif1 and RPA, we performed the Co-IP experiments, now described in Figure EV2. No decrease of interaction between Pif1-*pip* and RPA was detected relative to the WT control.

The *rbm* mutation is not expected to affect the interaction between Pif1 and PCNA based on the data published in Buzovetsky et al., 2017 (PMID: 29141206). In this publication, the authors have shown that the N-terminus of Pif1 (spanning the RBM motif) is not required for Pif1 to interact with PCNA *in vitro*.

- Figure 1A needs a loading control (e.g. anti-GST western blot)

We agree with this suggestion. The experiments in Figures 1A and 1D were repeated with the controls and supplemented with the loading control blots (anti-GST western blots).

Minor points:

- Is the RPA-binding motif evolutionarily conserved in Pif1-family helicases from other eukaryotes?

The N-terminus of *S. cerevisiae* Pif1 is mostly unique to the *Saccharomyces* genus, however the motifs similar to the RBM motif can be found within the *Saccharomycetaceae* family (see the alignment below). This leaves the possibility that Pif1 in other eukaryotes contain the RPA-interacting motifs within their unstructured N- or C-termini or evolved to be recruited via alternative recruitment pathways.

Pif1	SRGFRSNNFIQAQLKHP	SILSKEDLDLLSDSD	DWEIPDCIQLE
Z.rouxii	SSLSQSSNSSKRLKLSH	PEYDELDKLLSDSD	GWEDIAEVKLLK
T.delbrueckii	SSVQRTLKTSRKEITHT	PRNDYDELNDLLSDSD	GWEDDIHVERV
K.africana	SSLESTPINANKKPKLTS	HELSELADMLSDSD	DWEGNIQPNVH
N.castellii	SKSPSWMVQSEGNIKRQ	KLSEELNILLSDQ	DDWMIYMSDNIV

Dear Dr. Makovets,

Thank you for submitting your revised manuscript. It has now been seen by two of the original referees.

As you can see, the referees find that the study is significantly improved during revision and recommend publication. However, I need you to address the points below before I can accept the manuscript.

- Please rename the Competing Interests section as "Disclosure Statement and Competing Interests".
- Please remove the Author Contributions section from the manuscript.
- Tables EV1-EV3 should be uploaded as individual files, one per table, in word, excel or PDF formats. Their legends need to be in the files, not in the manuscript text.
- The manuscript sections should be in the following order: Title page - Abstract & Keywords - Introduction - Results - Discussion - Materials & Methods - Data Availability - Acknowledgments - Disclosure Statement & Competing Interests - References - Figure Legends - Tables with legends - Expanded View Figure Legends.
- Our production/data editors have asked you to clarify several points in the figure legends:
 - o Although 'n' is provided, please describe the nature of entity for 'n' in the legends of figures 1b, e; 2a-c, e; 3c, e-f; EV 1b; EV 2b.
 - o Please note that the legend for figure 2f is mislabeled as 2d. This needs to be rectified.

Thank you again for giving us to consider your manuscript for EMBO Reports, I look forward to your minor revision.

Kind regards,

Deniz Senyilmaz Tiebe

--

Deniz Senyilmaz Tiebe, PhD
Editor
EMBO Reports

Referee #1:

I am satisfied with the changes made by the authors.

Referee #3:

The authors have successfully addressed all my comments from the first review. As it is, this manuscript will be a strong contribution to the genome stability field.

The authors have addressed all minor editorial requests.

Dear Dr. Makovets,

Thank you for submitting your revised manuscript. I have now looked at everything and all is fine. Therefore, I am very pleased to accept your manuscript for publication in EMBO Reports.

Congratulations on a nice work!

Kind regards,

Deniz Senyilmaz Tiebe

--

Deniz Senyilmaz Tiebe, PhD

Editor

EMBO Reports

--
